# Dataset Condensation with Sharpness-Aware Trajectory Matching

## Abstract

Dataset condensation aims to synthesise datasets with a few representative samples that can effectively represent the original datasets. This enables efficient training and produces models with performance close to those trained on the original sets. Most existing dataset condensation methods conduct dataset learning under the bilevel (inner and outer loop) based optimisation. However, due to its notoriously complicated loss landscape and expensive time-space complexity, the preceding methods either develop advanced training protocols so that the learned datasets generalise to unseen tasks or reduce the inner loop learning cost increasing proportionally to the unrolling steps. This phenomenon deteriorates when the datasets are learned via matching the trajectories of networks trained on the real and synthetic datasets with a long horizon inner loop. To address these issues, we introduce Sharpness-Aware Trajectory Matching (SATM), which enhances the generalisation capability of learned synthetic datasets by minimising sharpness in the outer loop of bilevel optimisation. Moreover, our approach is coupled with an efficient hypergradient approximation that is mathematically well-supported and straightforward to implement along with controllable computational overhead. Empirical evaluations of SATM demonstrate its effectiveness across various applications, including standard in-domain benchmarks and out-of-domain settings. Moreover, its easy-to-implement properties afford flexibility, allowing it to integrate with other advanced sharpness-aware minimisers. We will release our code on GitHub.

## 1 Introduction

The success of modern deep learning in various fields, exemplified by Segment Anything (Kirillov et al., 2023) in computer vision and GPT (Ouyang et al., 2022) in natural language processing, comes at a significant cost in terms of the enormous computational expenses associated with large-scale neural network training on massive amounts of real-world data Radford et al. (2021); Li et al. (2023); Schuhmann et al. (2022); Li et al. (2022); Gowda et al. (2023). To reduce training and dataset storage costs, selecting the representative subset based on the specific importance criteria forms a direct solution (Har-Peled & Mazumdar, 2004; Yang et al., 2022; Paul et al., 2021; Wang et al., 2022b). However, these methods fail to handle the cases when the samples are distinct and the information is uniformly distributed in the dataset. In contrast, Dataset Condensation (DC) (Zhao et al., 2021; Zhao & Bilen, 2023; Wang et al., 2018; Cazenavette et al., 2022; Du et al., 2023) focuses on creating a small, compact version of the original dataset that retains its representative qualities. Models trained on the condensed dataset perform comparably to those trained on the full dataset. This approach significantly reduces training costs and storage requirements, meanwhile expedites more challenging machine learning tasks such as hyperparameter tuning, continual learning (Rosasco et al., 2021), architecture search (Sangermano et al., 2022; Yu et al., 2020; Masarczyk & Tautkute, 2020), and privacy-preserving (Shokri & Shmatikov, 2015; Dong et al., 2022).

Given the significant practical value of condensed datasets, considerable effort has been directed toward designing innovative surrogate methods to ensure that synthetic datasets capture representative signals, thereby enhancing future deployments' performance (Zhao & Bilen, 2023; Zhao et al., 2021; Zhou et al., 2022; Kim et al., 2022). Bilevel Optimisation (BO) provides a DC paradigm learning synthetic dataset through its main optimisation objective in the outer loop constrained by training neural networks in its inner loop. One line of the representative solutions condenses datasets

by minimising the disparity between training trajectories on synthetic and real sets, achieving notable performance (Cazenavette et al., 2022). The following studies either reduce the computational cost of inner loop unrolling or steer the optimisation process to enhance the generalization of the learned dataset to the unseen tasks. For instance, FTD (Du et al., 2023) improves the performance of synthetic datasets by leveraging high-quality inner loop expert trajectories and incorporating momentum into the outer loop optimisation via Exponential Moving Average (EMA) and the introduced memory overhead increases along with the synthetic dataset budget. TESLA (Cui et al., 2023) is proposed with a two-inner loop-based algorithm to approximate the hypergradient for the dataset updates maintaining a constant memory usage. In this work, we introduce a two inner loop-based mechanism, that directly optimizes the generalisation ability of the synthetic dataset with controllable memory cost. This results in superior performance on both in-domain and out-of-domain tasks, with reduced memory and time complexity compared to those methods.

Inspired by (Foret et al., 2020; Kwon et al., 2021; Li & Giannakis, 2024), the studies on improving generalisation by minimising loss landscape sharpness to achieve flat convergence regions in uni-level optimisation, we extend this concept and develop an algorithm for dataset condensation in the more complex bilevel optimisation setting. Our approach addresses the computational overhead caused by the notorious ascent and descent routine in sharpness-aware optimisers, which typically double both the time and memory costs throughout the learning process. Specifically, we propose a lightweight and efficient trajectory matching-based method, Sharpness-Aware Trajectory Matching (SATM), that enhances the generalisation of the alliance trajectory matching algorithm significantly and integrates with the beneficial properties introduced by FTD (Du et al., 2023) and TESLA Cui et al. (2023) with noticeable improvement margin across various applications whilst avoiding the redundant computation graph holding and recomputing. The main contributions of this work are summarised as:

- We primarily study the generalisation ability of the outer loop in the bilevel optimisation for the learned dataset, then design an algorithm, Sharpness-Aware Trajectory Matching, to jointly minimise the sharpness and the distance between training trajectories with a tailored loss landscape smoothing strategy.

- A simple and easy-to-implement method is proposed to handle the tremendous computational overhead introduced by the sharpness proxy in the long inner loop horizons scenario. In addition, to reduce the redundancy of the (hyper) gradient calculation, the learning rate in the inner loop is learned by simple model gradient aggregation without holding the computational graph.

- We provide rigorous theoretical support for the proposed approximation methods by bounding the errors of the approximations and analysing the approximation error effected by the hyperparameters, which shed light on meaningful hyperparameter tuning.

- SATM outperforms the trajectory-matching-based competitors on various condensation benchmarks with noticeable margins on in- and out-of-domain settings.

## 2 BACKGROUND AND RELATED WORK

### 2.1 BILEVEL OPTIMISATION AND DATASET CONDENSATION

Bilevel Optimisation (Sinha et al., 2017; Zhang et al., 2024), nesting optimisation problems as constraints for the main optimisation objective, is formulated as follows:

$$\min_{\phi} \mathcal{L}^{outer}(\theta^*(\phi), \phi) \tag{1}$$

$$\textbf{s.t. } \theta^*(\phi) = \arg\min_{\theta} \mathcal{L}^{inner}(\theta, \phi) \tag{2}$$

where, $\arg\min_{\theta} \mathcal{L}^{inner}(\theta, \phi)$ forms the constraint for the main optimisation objective function, $\mathcal{L}^{outer}$. The learnable parameter $\phi$ in the outer loop influences the performance of the inner loop state, $\theta(\phi)$, while the inner loop also depends on the current free parameter on the outer loop. This optimisation framework is widely used in various machine learning areas, including hyperparameter tuning (Lorraine et al., 2020; Maclaurin et al., 2015; MacKay et al., 2019) and meta-learning (Finn et al., 2017; Gao et al., 2022; Rajeswaran et al., 2019; Gao et al., 2021).

Inspired by knowledge distillation (Gou et al., 2021; Yang et al., 2020), Wang *et al.* (Wang et al., 2018) leverage BO to distill a small, compact synthetic dataset for efficient training on unseen downstream tasks. Several works expanding on this BO framework match gradients (Zhao & Bilen, 2021; Zhao et al., 2021; Lee et al., 2022), features (Wang et al., 2022a), and distributions (Zhao & Bilen, 2023) produced by the synthetic and real sets. They achieve this with a few iterations of inner loop unrolling to avoid the challenges of nested optimisation. To address the same challenge, Nguyen *et al.* (Nguyen et al., 2021b;a) directly estimate the convergence of the inner loop using the Neural Tangent Kernel (NTK) to emulate the effects from the synthetic sets. However, due to the heavy computational demands of matrix inversion, the NTK-based method struggles to scale up for condensing large, complex datasets. MTT (Cazenavette et al., 2022) emphasises the benefits of a long horizon inner loop and minimises the differences between synthetic and expert training trajectory segments. Nonetheless, the learned synthetic dataset often overfits the neural architecture used in the expert trajectories, resulting in limited generalisation ability. In this work, we address this problem by exploring the flatness of the synthetic dataset's convergence region.

## 2.2 SHARPNESS-AWARE MINIMISATION

The generalisation enhanced by flat region minimums has been observed empirically and studied theoretically (Dinh et al., 2017; Keskar et al., 2016; Neyshabur et al., 2017). Motivated by this, Sharpness-aware minimiser (SAM) (Foret et al., 2020) optimises the objective function and sharpness simultaneously to seek the optimum lying in a flat convergence region. Given the training data, $D$, consider a training problem where the objective function is denoted as $\mathcal{L}(\phi; D)$ with the learnable parameter $\phi$, the objective function of SAM is framed as:

$$\min_{\phi} \max_{||\epsilon||_2 \leq \rho} \mathcal{L}(\phi + \epsilon; D) \tag{3}$$

where approximating sharpness is achieved by finding the perturbation vectors $\epsilon$ maximising the objective function in the Euclidean ball with radius, $\rho$, with the sharpness defined as:

$$\max_{||\epsilon||_2 \leq \rho} \left| \mathcal{L}(\phi + \epsilon; D) - \mathcal{L}(\phi; D) \right|. \tag{4}$$

Instead of solving this problem iteratively, a closed-form approximation of the optimal by utilisation of the first-order Taylor expansion of the training loss is given by

$$\epsilon = \rho \frac{\nabla \mathcal{L}(\phi)}{||\nabla \mathcal{L}(\phi)||_p} \approx \arg\max_{||\epsilon|| \leq \rho} \mathcal{L}(\phi + \epsilon),$$

Overall, the updating procedure of SAM in each iteration is summarised as follows:

$$\phi = \phi - \alpha \nabla \mathcal{L}(\phi + \epsilon) \quad \textbf{s.t.} \ \ \epsilon = \rho \frac{\nabla \mathcal{L}(\phi)}{||\nabla \mathcal{L}(\phi)||_p} \tag{5}$$

where $\alpha$ represents the learning rate and after computing the gradient, $\nabla \mathcal{L}(\phi + \epsilon)$, the parameter update procedure is instantiated by standard optimisers, such as SGD and Adam (Kingma & Ba, 2015). Without losing generality, we set $p = 2$ for simplicity for the rest of this work. One can observe that due to the two-stage gradient calculation at $\phi$ and $\phi + \epsilon$, the computational overhead of SAM is double, compared with the conventional optimisation strategy. To reduce the computational cost, ESAM (Du et al., 2022) randomly selects a subset of the parameters to update in each iteration. Zhuang *et al.* (Zhuang et al., 2021) observes that SAM fails to identify the sharpness and mitigates this by proposing a novel sharpness proxy. To tackle the complicated loss landscape, Li and Giannakis (Li & Giannakis, 2024) introduce a momentum-like strategy for sharpness approximation while ASAM (Kwon et al., 2021) automatically modify the sharpness reaching range by adapting the local loss landscape geometry. In contrast, we handle complicated multi-iteration unrolling for learning datasets in the many-shot region where both the difficulty of approximating the sharpness and the computation resources surge.

## 3 METHOD

We introduce our method in this section starting with reviewing a DC framework, Matching Training Trajectory (MTT) (Cazenavette et al., 2022), applied in this work. Then we combine the bilevel optimisation with sharpness-aware optimisation tailored for dataset condensation with a loss landscape

---

Algorithm 1: Sharpness-Aware Trajectory Matching for dataset condensation.

1: **Input:** $\{\theta_t^E\}_0^T, \alpha, \beta$.
2: **Output:** $\phi$
3: Init $\phi$
4: **while** not converged or reached max steps **do**
5:    Sample an iteration $t$ to construct an expert segment, $\theta_t^E$, and $\theta_{t+M}^E$
6:    $\theta^S = \theta_t^E$
7:    $\phi_j^\Delta \sim \mathcal{N}(0, \gamma||\phi_j||_2 I)$
8:    $\phi = \phi + \phi^\Delta$
9:    **for all** $i \leftarrow 1$ to $N$ **do**
10:      $\theta^S = \theta^S - \alpha\nabla\mathcal{L}(\theta^S, \phi)$
11:    **end for**
12:    Compute $\nabla F(\phi)$ by Eq. 10
13:    $\epsilon = \rho\nabla F(\phi)/||\nabla F(\phi)||_2$
14:    $\bar{\theta}^S = \theta_{t+\kappa}^S$
15:    $\phi = \phi - \phi_\Delta$
16:    **for all** $i \leftarrow N - \tau$ to $N$ **do**
17:      $\bar{\theta}^S = \bar{\theta}^S - \alpha\nabla\mathcal{L}(\bar{\theta}^S, \phi + \epsilon)$
18:    **end for**
19:    Compute $\nabla F(\phi + \epsilon)$ by Eq. 11
20:    $\phi = \phi - \beta\nabla F(\phi + \epsilon)$
21: **end while**

---

smoothing strategy for accurate sharpness approximation. To efficiently reduce the computational burden introduced by the sharpness-aware minimisers, we design and analyse time and memory-saving hypergradient approximations for the long horizon inner loop with the general method outlined in Algorithm 1.

### 3.1 PRELIMINARY

With the assumption that the datasets containing similar information generate close training trajectories, MTT (Cazenavette et al., 2022) proposed to create the synthetic datasets by minimising the distance between the training trajectory produced by the synthetic set, named synthetic trajectories, and those by the real set, termed expert trajectories. A sequence of expert weight checkpoints, $\theta_t^E$, are collected during the training on the real sets in the order of iterations, $t$, to construct the expert trajectories, $\{\theta_t^E\}_{t=0}^T$ with T denoting the total length of the trajectory. The pipeline of MTT starts with sampling a segment of expert trajectory, starting from $\theta_t^E$ to $\theta_{t+M}^E$ with $0 \leq t \leq t + M \leq T$. Then, to generate a synthetic segment, a model, $\theta_t^S$, is initialised by, $\theta_t^E$, and trained on the learnable dataset, $\phi$, to get $\theta_{t+N}^S(\phi)$ after $N$ iteration. Afterwards, the disparity between $\theta_{t+N}^S(\phi)$ and $\theta_{t+M}^E$ is optimised to learn the synthetic dataset. Formally, the dataset condensation algorithm can be described as:

$$\min_\phi \mathcal{L}(\theta^S(\phi)) := \frac{1}{\delta}||\theta_{t+N}^S(\phi) - \theta_{t+M}^E||_2^2 \tag{6}$$

$$\text{s.t. } \theta_{t+N}^S(\phi) = \Xi_N(\theta_t^S, \phi)$$

where $\Xi_N(\cdot)$ represents N differentiable minimising steps on the inner loop objective, CrossEntropy loss, $\mathcal{L}_{CE}(\theta, \phi)$. The existing optimisers can instantiate this operation, such as SGD whose one-step optimisation is exemplified by $\Xi(\theta, \phi) = \theta - \alpha\nabla\mathcal{L}_{CE}(\theta, \phi)$ where $\alpha$ denotes the learning rate. Note M and N are not necessarily equal since dense information in the synthetic datasets leads to fast training. $\delta$, stabilising the numerical computation, can be unpacked as $||\theta_t^E - \theta_{t+M}^E||_2^2$.

### 3.2 SMOOTH SHARPNESS-AWARE MINIMISATION IN OUTER LOOP

Generalising to the unseen tasks is challenging for the learned synthetic datasets. To mitigate this issue, we steer the optimisation on the outer loop in Eq. 6 and minimise the objective function forward landing in the flat loss landscape region to enable the synthetic data to be generalised to both in- and out-of-domain settings. This property has been studied in (Petzka et al., 2021; Kaddour et al., 2022), in the uni-level optimisation. In this work, we forage this into the bilevel optimisation

framework by integrating Shaprness-Aware minimisation. To jointly optimise the sharpness of the outer loop and the distance between the trajectory w.r.t to the synthetic dataset, we maximise the objective function in the $\rho$ regime for the sharpness proxy approximation and then optimise the distance between trajectories according to the gradient computed on the local maximum for the dataset learning. This process is described as:

$$\min_{\phi} \max_{||\epsilon||_2 \leq \rho} \mathcal{L}(\theta^S(\phi + \epsilon)) = \frac{1}{\delta}||\theta^S_{t+N}(\phi + \epsilon) - \theta^E_{t+M}||^2_2 \tag{7}$$

$$\text{s.t. } \theta^S_{t+N}(\phi) = \Xi_N(\theta^S_t, \phi). \tag{8}$$

We define $F(\phi) = \mathcal{L}(\theta^S_{t+N}(\phi))$ to eliminate the effect of the inner loop solution on the outer loop loss value without losing generality. The perturbation vector, $\epsilon$, is computed through a closed-form solution derived through the first-order Taylor expansion of the objective function in Eq. 6.

$$\begin{aligned}
\epsilon = \arg\max_{||\epsilon||_2 \leq \rho} \mathcal{L}(\theta^S(\phi + \epsilon)) &= \arg\max_{||\epsilon||_2 \leq \rho} F(\phi + \epsilon) \\
&\approx \arg\max_{||\epsilon||_2 \leq \rho} F(\phi) + \epsilon \cdot \nabla F(\phi) \\
&= \arg\max_{||\epsilon||_2 \leq \rho} \epsilon \cdot \nabla F(\phi) \approx \rho \frac{\nabla F(\phi)}{||\nabla F(\phi)||_2}.
\end{aligned} \tag{9}$$

The closed-form solution given in Eq. 9 can be interpreted as a one-step gradient ascent. However, this one-step gradient ascent may fail to reach the local maximum of the sharpness proxy, due to the high variance of hypergradient caused by the complicated outer loop loss landscape. This phenomenon has also been observed by (Liu et al., 2022; Du et al., 2022) in the uni-level optimisation and will aggravate in the complicated bilevel case (Abbas et al., 2022). To conduct accurate sharpness approximation, motivated by (Liu et al., 2022; Haruki et al., 2019; Wen et al., 2018; Duchi et al., 2012), we introduce fluctuation on the learnable dataset to smooth the landscape. To be more specific, each synthetic image indexed by $j$ is perturbed by a random noise sampled from a Gaussian distribution with a diagonal covariance matrix whose magnitude is proportional to the norm of each image $||\phi_j||$:

$$\phi_j = \phi_j + \phi_j^\Delta, \quad \phi_j^\Delta \sim \mathcal{N}(0, \gamma||\phi_j||_2)$$

where $\gamma$ is a tunable hyperparameter controlling the fluctuation strength. This process is conducted on the image independently in each one-step gradient ascent.

### 3.3 EFFICIENT SHARPNESS-AWARE MINIMISATION IN BILEVEL OPTIMISATION

One can notice that a one-step update in the outer loop needs to compute the hypergradient twice with one for the perturbation vector $\epsilon$ and the other for the real update gradient, $\nabla F(\phi)$. Directly computing those two gradients will double the computation cost in contrast with MTT and FTD instead of TESLA which we will discuss later. To alleviate this problem, we proposed two approximation strategies, Truncated Unrolling Hypergradient (TUH) and Trajectory Reusing (TR).

#### 3.3.1 TRUNCATED UNROLLING HYPERGRADIENT

The long inner loop horizon introduces tremendous computational overhead. In our dataset condensation framework, the hypergradient for updating the learnable dataset is computed by differentiating through the unrolled computational graph of the inner loop. This vanilla hypergradient computation lets the memory cost scale with the number of the inner loop iterations which is not feasible as condensing the complicated datasets requires long horizon inner loops. Instead, we *truncate the backpropagation* by only differentiating through the last several steps of the inner loop. This reduces both the required memory and computational time. More concretely, the truncated hypergradient computation with $N$ step unrolling can be expressed as:

$$\frac{\partial F_\iota(\phi)}{\partial \phi} = \frac{\partial \mathcal{L}(\theta(\phi))}{\partial \theta_\iota}\frac{\partial \theta_\iota}{\partial \phi} = \sum_{i=\iota}^{N} \frac{\partial \mathcal{L}(\theta(\phi))}{\partial \theta_N} \left( \prod_{i'=i}^{N} \frac{\partial \theta_{i'}}{\partial \theta_{i'-1}} \right) \frac{\partial \theta_i}{\partial \phi}, \tag{10}$$

where $\iota$ controls the number of truncated steps that $N - \iota$ steps of the inner loop will be differentiated through. In addition, the risk of hyerpgradient exploding and vanishing caused by the ill-Jabian

$\frac{\partial \theta_i}{\partial \theta_{i-1}}$, which may happen in any inner loop step, can be reduced. This mechanism can be easily implemented by unholding the computational graph while optimising the inner loop and then creating the computational graph at a certain iteration with Pytorch-based pseudocode given in Appx. A.2.

Following (Shaban et al., 2019; Bolte et al., 2024), we analyse the discrepancy between hypergradients computed by the truncated and untruncated computational graph in the setting where the synthetic trajectory is produced by optimised from the initialisation $\theta_0^E$ until converge.

**Proposition 3.1.** *Assmue $\mathcal{L}_{CE}$ is $K$-smooth, twice differentiable, and locally $J$-strongly convex in $\theta$ around $\{\theta_{\iota+1}, ..., \theta_N\}$. Let $\Xi(\theta, \phi) = \theta - \alpha \nabla \mathcal{L}_{CE}(\theta, \phi)$. For $\alpha \leq \frac{1}{K}$, then*

$$\left\| \frac{\partial F(\phi)}{\partial \phi} - \frac{\partial F_\iota(\phi)}{\partial \phi} \right\| \leq 2^\iota (1 - \alpha J)^{N-\iota+1} \left\| \frac{\partial \mathcal{L}(\theta(\phi))}{\partial \theta_N(\phi)} \right\| \max_{i \in \{0,..\iota\}} \left\| \frac{\partial \theta_i}{\partial \phi} \right\|$$

*where $\frac{\partial F(\phi)}{\partial \phi}$ denotes the untruncated hypergradient.*

The Proposition 3.1 shows that the error of the truncated hypergradient decreases exponentially in $N - \iota + 1$ when $\theta$ converges to the neighbourhood of a local minimum in the inner loop and the proof is given in Appx. A.3.

### 3.3.2 TRAJECTORY REUSING

The sharpness-aware minimisation requires computing the gradient twice for sharpness proxy approximation and free parameter update, which means in bilevel optimisation the inner loop is required to unroll twice. This boosts the computational spending and slows down the training speed when inner loops comprise long trajectories. To improve the efficiency of training by benefiting from the existing knowledge, we propose to reuse the trajectory generated by the first round of inner loop unrolling. We denote the trajectories generated by training on the perturbed dataset as $\hat{\theta}_i(\phi+\epsilon)$. Other than unrolling the entire second trajectory initialised by the expert segment, the training is initialised by the middle point, indexed by $\tau$, from the first trajectory $\hat{\theta}_\tau(\phi+\epsilon) := \theta_\tau(\phi)$. Note that the hypergradient for the dataset update is truncated implicitly since this hypergradient approximation will not consider the steps earlier than $\tau$ which is further constrained, $\tau \geq \iota$. Coupled with the same truncated strategy for the first round, the hypergradient in the second trajectory is computed as:

$$\frac{\partial F_{\tau,\epsilon}(\phi)}{\partial \phi} = \frac{\partial \mathcal{L}(\theta(\phi))}{\partial \theta_\tau} \frac{\partial \theta_\tau}{\partial \phi} = \sum_{i=\tau}^{N} \frac{\partial \mathcal{L}(\theta(\phi))}{\partial \theta_N} \left( \prod_{i'=i}^{N} \frac{\partial \theta_{i'}}{\partial \theta_{i'-1}} \right) \frac{\partial \theta_i}{\partial \phi} \Bigg|_{\phi=\phi+\epsilon,\ \hat{\theta}_\tau(\phi+\epsilon)=\theta_\tau(\phi)} \tag{11}$$

One may notice that the trajectory reusing strategy assumes the difference between two trajectories before step $\tau$ can be ignored. To rigorously study the effect of this assumption, we analyse the distance between $\theta_\tau(\phi)$ and $\theta_\tau(\phi + \epsilon)$. Similar to the Growth recursion lemma (Hardt et al., 2016) applied to upper-bound the difference between two weight points of two different trajectories trained by the dataset with only one data point difference. We develop the bound for the difference between two weight points at the same iteration of their trajectories generated by the datasets with and without perturbation below. The proof is provided in Appx.A.1.

**Theorem 3.2.** *Let $\mathcal{L}(\phi, \theta)$ be a function that is $\sigma$-smooth and continuous with respect to its arguments $\phi$ and $\theta$. Additionally, let the second-order derivatives $\nabla_\phi \nabla_\theta \mathcal{L}(\phi, \theta)$ be $\beta$-continuous. Consider two trajectories obtained by conducting gradient descent training on the datasets $\phi$ and $\phi+\epsilon$, respectively, with a carefully chosen learning rate $\alpha$ and identical initializations. After $\tau$ steps of training, let $\Delta\theta_\tau = \hat{\theta}_\tau(\phi + \epsilon) - \theta_\tau(\phi)$. Then, we have:*

$$\|\Delta\theta_\tau\| \leq \alpha\tau(2\sigma + \beta\rho).$$

This theorem tells us that the bound of the distance of those two points is associated with the learning rate and the number of iterations. Thus, when the learning rate and $\tau$ are selected reasonably, $\theta_\tau(\phi)$ approximate $\hat{\theta}_\tau(\phi + \epsilon)$ properly. In addition, we set $\tau = \iota$ in our experiments to reduce the hyperparameter tuning efforts even though tuning them separately may achieve better results. We compare the time and memory complexity of our method and Reverse Model Reverse Mode Differentiation (RMD) used in MTT (Cazenavette et al., 2022) and FTD (Du et al., 2023) in Table 1 to exhibit the efficiency provided by our method.

| Methods | Time | Memory |
|---|---|---|
| MTT, FTD (RMD) | $\mathcal{O}(cN)$ | $\mathcal{O}(PN)$ |
| TESLA | $\mathcal{O}(2cN)$ | $\mathcal{O}(P)$ |
| TUH + TR | $\mathcal{O}(cN + c\tau)$ | $\mathcal{O}(P(N - \iota))$ |

Table 1: The computational complexity comparison for different trajectory matching based algorithms in time and memory cost. $c$ is the time cost for computing $\Xi(\theta, \phi)$ with $\theta \in R^P$ and $\phi \in R^Q$. P and Q denote the dimensions of the base model and synthetic dataset.

**Learning-Rate Learning with First Order Derivative:** Adapting the inner loop learning rate, $\alpha$, to the different stages of dataset learning determines the performance of the learned dataset (Cazenavette et al., 2022). The automatic adaption is achieved by modifying the learning rate by the hypergradient of the dataset learning objective function, $\frac{\partial \mathcal{L}(\phi)}{\partial \alpha}$. This hypergradient can be computed jointly with the hypergradient for the dataset learning which is cumbersome in practice. To mitigate this burden, we derive an analytic solution for inner loop learning rate updating:

$$\alpha = \alpha - \lambda \frac{\partial \mathcal{L}(\theta_N(\phi))}{\partial \theta_N} \cdot \left( - \sum_{i=0}^{N-1} \frac{\partial \mathcal{L}_{CE}(\theta_i, \phi)}{\partial \theta_i} \right) \tag{12}$$

where $\lambda$ indicates the learning rate for the learning rate learning and the derivation given in Appx. A.4. This closed-form solution only aggregates the gradient of each step instead of differentiating through the inner loop unrolling graph, simplifying the hypergradient computation. As can be noticed, two inner loop trajectories in the sharpness aware setting are capable of this Eq. 12. We chose the first in our experiments due to the implementation simplicity without causing any significant performance differences. The visualisation comparison of the learning rate learning dynamic produced by the first and second-order derivative is illustrated in Fig. 1.

In essence, SATM is designed to conduct efficient sharpness minimisation in the outer loop of the bilevel optimisation-based dataset condensation methods and the proposed efficiency strategies, including THU and TR, are flexible enough to adapt to other advanced sharpness-aware optimisers such as ASAM (Kwon et al., 2021) and Vasson (Li & Giannakis, 2024).

## 4 EXPERIMENTS

We evaluate SATM on various in-domain tasks where the neural architecture and data distribution on the dataset learning and test stage are the same with different datasets and different numbers of images per category (IPC) . Besides, cross-architecture and cross-task evaluation are conducted to demonstrate the generalisation achieved in sharpness minimisation on out-of-domain settings.

### 4.1 EXPERIMENTS SETTINGS

**Dataset:** We conduct experiments on three main image datasets, Cifar10 (Krizhevsky et al., 2009), Cifar100 (Krizhevsky et al., 2009) and TinyImageNet (Le & Yang, 2015). Cifar10 categorises 50,000 images with the size $32 \times 32$ into 10 classes while Cifar100 further categorises each of those 10 classes into 10 fine-grained subcategories. TinyImageNet comprises 100,000 images distributed across 200 categories, each category consisting of 500 images resized to dimensions of $64 \times 64$. We further evaluate SATM on the subset of ImageNet, namely ImageNette, Image Woof, ImageFruit and ImageMeow with each set containing 10 different categories of $128 \times 128$ images.

**Training and Evaluation:** The expert trajectories for Cifar10 and Cifar100 are trained with 3-layer ConvNet and collected after each epoch with the initialisation, and those for TinyImageNet and ImageNet are trained with 4-layer and 5-layer ConvNet Gidaris & Komodakis (2018) respectively. In the in-domain setting, the synthetic datasets are learned and evaluated on the same architectures while in the out-of-domain settings, the learned synthetic datasets are deployed to train different architectures, such as AlexNet (Krizhevsky et al., 2012), VGG11 (Simonyan & Zisserman, 2014) and ResNet18 (He et al., 2016), which is novel to the synthetic datasets. The trained neural networks are evaluated on the real test sets for generalisation ability comparison of the synthetic datasets.

| Method | IPC | DC | DSA | DM | MTT | FTD | TESLA | MDC | Ours |
|---|---|---|---|---|---|---|---|---|---|
| Cifar-10 | 1 | $28.3_{\pm0.5}$ | $28.8_{\pm0.7}$ | $26.0_{\pm0.8}$ | $46.2_{\pm0.8}$ | $46.8_{\pm0.3}$ | $48.5_{\pm0.8}$ | $47.5_{\pm0.4}$ | $\mathbf{49.0}_{\pm0.3}$ |
| | 3 | - | - | - | $55.3_{\pm0.4}$ | $56.0_{\pm0.2}$ | - | $56.0_{\pm0.3}$ | $\mathbf{57.1}_{\pm0.4}$ |
| | 10 | $44.9_{\pm0.5}$ | $52.1_{\pm0.6}$ | $48.9_{\pm0.6}$ | $65.4_{\pm0.7}$ | $66.6_{\pm0.3}$ | $66.4_{\pm0.8}$ | $66.7_{\pm0.7}$ | $\mathbf{67.1}_{\pm0.3}$ |
| | 50 | $53.9_{\pm0.5}$ | $60.6_{\pm0.5}$ | $63.0_{\pm0.4}$ | $71.6_{\pm0.2}$ | $73.8_{\pm0.3}$ | $72.6_{\pm0.7}$ | $73.7_{\pm0.3}$ | $\mathbf{73.9}_{\pm0.2}$ |
| Cifar-100 | 1 | $12.8_{\pm0.3}$ | $13.9_{\pm0.3}$ | $11.4_{\pm0.3}$ | $24.3_{\pm0.3}$ | $25.2_{\pm0.2}$ | $24.8_{\pm0.4}$ | $25.9_{\pm0.2}$ | $\mathbf{26.1}_{\pm0.4}$ |
| | 3 | - | - | - | $32.6_{\pm0.4}$ | $33.1_{\pm0.4}$ | - | $33.3_{\pm0.3}$ | $\mathbf{33.9}_{\pm0.2}$ |
| | 10 | $25.2_{\pm0.3}$ | $32.3_{\pm0.3}$ | $29.7_{\pm0.3}$ | $39.7_{\pm0.4}$ | $\mathbf{43.4}_{\pm0.3}$ | $41.7_{\pm0.3}$ | $42.7_{\pm0.6}$ | $43.1_{\pm0.5}$ |
| | 50 | - | $42.8_{\pm0.4}$ | $43.6_{\pm0.4}$ | $47.7_{\pm0.2}$ | $50.7_{\pm0.3}$ | $47.9_{\pm0.3}$ | $49.6_{\pm0.4}$ | $\mathbf{50.9}_{\pm0.5}$ |
| TinyImageNet | 1 | - | - | $3.9_{\pm0.2}$ | $8.8_{\pm0.3}$ | $10.4_{\pm0.3}$ | - | $9.9_{\pm0.2}$ | $\mathbf{10.9}_{\pm0.2}$ |
| | 3 | - | - | - | $10.5_{\pm0.3}$ | $11.6_{\pm0.5}$ | - | $12.4_{\pm0.3}$ | $\mathbf{13.6}_{\pm0.4}$ |
| | 10 | - | - | $12.9_{\pm0.4}$ | $23.2_{\pm0.2}$ | $24.5_{\pm0.2}$ | - | $24.8_{\pm0.4}$ | $\mathbf{25.4}_{\pm0.4}$ |

Table 2: Test Accuracy (%) Comparison of different image per category (IPC) setting on Cifar10, Cifar-100 and Tiny ImageNet: the models are trained on the syntactic dataset learned by MTT and our method independently and evaluated on the corresponding test set with real images. We cite the results of DC, DM and MMT from FTD (Du et al., 2023).

| | ImageNette | ImageWoof | ImageFruit | ImageMeow |
|---|---|---|---|---|
| MTT | $63.0_{\pm1.3}$ | $35.8_{\pm1.8}$ | $40.3_{\pm1.3}$ | $40.4_{\pm2.2}$ |
| FTD | $67.7_{\pm0.7}$ | $38.8_{\pm1.4}$ | $44.9_{\pm1.5}$ | $43.3_{\pm0.6}$ |
| Ours | $\mathbf{68.2}_{\pm0.5}$ | $\mathbf{39.4}_{\pm1.2}$ | $\mathbf{45.2}_{\pm1.3}$ | $\mathbf{45.4}_{\pm0.9}$ |
| All | $87.4_{\pm1.0}$ | $67.0_{\pm1.3}$ | $63.9_{\pm2.0}$ | $66.7_{\pm1.1}$ |

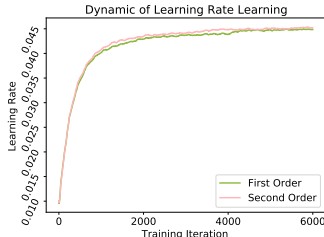

Table 3: Test accuracy (%) comparison on Cifar10 with 10 and 50 images per class setting: the syntactic datasets by MTT, FTD and our algorithm are learned on ConvNet and tested on AlexNet, VGG11 and ResNet18.

Figure 1: The comparison of the learning dynamic of learning rate learning with first and second order differentiation when condensing on the Cifar100-10IPC setting.

## 4.2 PRIMARY RESULTS

### 4.2.1 STANDARD DATASET CONDENSATION BENCHMARK

We compare our method against the other dataset condensation techniques, such as DC (Zhao et al., 2021), DSA (Zhao & Bilen, 2021), DM (Zhao & Bilen, 2023), MTT(Cazenavette et al., 2022), FTD (Du et al., 2023), TESLA (Cui et al., 2023) and MDC (He et al., 2024). The results from Table 2 demonstrate the benefits of the flat minima that SATM outperforms the competitors on almost all the settings of the standard dataset condensation benchmarks with various of IPCs. This benefit can be further observed in the high-resolution image condensation task in Table 3. Note that in our case, we merely build SATM up on Vanilla MMT (Cazenavette et al., 2022) without integrating the flat trajectory trick in FTD and the soft label in TESLA. Limited by the computational resource, we cannot conduct full batch training on Cifar100 with 10 IPC, 50 IPC and Tiny ImageNet with 10 IPC as that utilised on MTT and FTD, which we believe is the main reason that SATM performs slightly worse than FTD on the Cifar100 with 10 IPC setting. Besides, there are clear improvement margins over other trajectory-matching-based DC competitors. Moreover, in this work, we are also interested in studying whether the advantages brought by the flatness can also be observed in cross-architecture tasks, which leads to numerous practical applications. In Table 5, the synthetic datasets by learned SATM for Cifar10 exhibit strong generalisation ability across the unseen architectures on both IPC 10 and 50 settings over the candidate architectures in comparison with those learned by MTT (Cazenavette et al., 2022), FTD (Du et al., 2023) and TESLA (Cui et al., 2023). Additionally, one can notice that the performance of the learned dataset from the in-domain setting is not guaranteed in the cross-architecture setting. For instance, FTD performs similarly to SATM in the Cifar10 with 10 and 50 IPC settings when deploying on ConvNet in the dataset learning stage. However, the performance gaps become remarkable once the same datasets are used across architectures.

| Dataset (IPC) | MTT | EMA | SAM | GSAM | ASAM | Vasso | SATM |
|---|---|---|---|---|---|---|---|
| Cifar100 (1) | $24.3_{\pm0.4}$ | $24.7_{\pm0.2}$ | $25.7_{\pm0.3}$ | $25.9_{\pm0.3}$ | $25.7_{\pm0.3}$ | $25.9_{\pm0.2}$ | $\mathbf{26.1}_{\pm0.3}$ |
| Tiny ImageNet (3) | $10.5_{\pm0.3}$ | $10.9_{\pm0.3}$ | $12.3_{\pm0.2}$ | $13.1_{\pm0.2}$ | $12.8_{\pm0.4}$ | $12.2_{\pm0.2}$ | $\mathbf{13.6}_{\pm0.2}$ |

Table 4: Test Accuracy (%) Comparison with the advanced sharpness aware minimisation methods including EMA, SAM, GSAM, ASAM and Vasso with the same expert trajectories as MTT.

| Methods | IPC | ConvNet | AlexNet | VGG11 | ResNet18 |
|---|---|---|---|---|---|
| MTT |  | $64.3_{\pm0.7}$ | $34.2_{\pm2.6}$ | $50.3_{\pm0.8}$ | $46.4_{\pm0.6}$ |
| FTD | 10 | $66.6_{\pm0.4}$ | $36.5_{\pm1.1}$ | $50.8_{\pm0.3}$ | $46.2_{\pm0.7}$ |
| Ours |  | $\mathbf{67.1}_{\pm0.5}$ | $\mathbf{37.8}_{\pm0.8}$ | $\mathbf{51.4}_{\pm0.3}$ | $\mathbf{47.7}_{\pm0.4}$ |
| MTT |  | $71.6_{\pm0.2}$ | $48.2_{\pm1.0}$ | $55.4_{\pm0.8}$ | $61.9_{\pm0.7}$ |
| FTD | 50 | $73.8_{\pm0.2}$ | $53.8_{\pm0.9}$ | $58.4_{\pm1.6}$ | $65.7_{\pm0.3}$ |
| Ours |  | $\mathbf{74.2}_{\pm0.3}$ | $\mathbf{56.9}_{\pm0.7}$ | $\mathbf{63.5}_{\pm1.1}$ | $\mathbf{66.1}_{\pm0.5}$ |

Table 5: Test accuracy (%) comparison on Cifar10 with 10 and 50 images per class setting: the syntactic datasets by MTT, FTD and our algorithm are learned on ConvNet and tested on AlexNet, VGG11 and ResNet18.

### 4.2.2 CONTINUAL LEARNING

We expose the learned dataset to the task incremental setting, following the same protocol discussed in Gdumb (Prabhu et al., 2020) for a fair comparison with datasets produced by competitors such as DM (Zhao & Bilen, 2023), MTT (Cazenavette et al., 2022), and FTD (Du et al., 2023). Typically, models encounter a sequence of data from different categories and lose access to data from previous categories after training. A limited memory budget is available to save dataset information from previous tasks, enabling models to retain gained knowledge while adapting to new tasks. In Figure 2, we show that at each stage, as new categories are received, our learned datasets consistently outperform others in three settings: 5-task incremental with 50 images per category on Cifar10, 10-and 20-task incremental with 3 IPC on Tiny ImageNet. Given the result in Fig 2, SATM consistently outperforms other methods whenever the models encounter new tasks on all the settings.

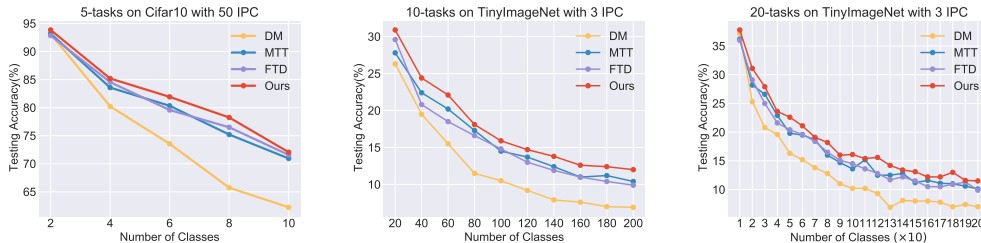

Figure 2: Test accuracy (%) comparison on continual learning. Left: 5-step class-incremental learning on Cifar10 50IPC, Middle: 10-step class-incremental learning on Tiny ImageNet 3IPC, Right: 20-step class-incremental learning on Tiny ImageNet 3IPC.

### 4.3 FURTHER ANALYSIS

### 4.3.1 COMPATIBILITY WITH ADVANCED SHARPNESS-AWARE OPTIMISERS

We study the compatibility of the proposed hypergradient approximation method on other sharpness minimisation-based methods including EMA, SAM (Foret et al., 2020), GSAM (Zhuang et al., 2021), ASAM (Kwon et al., 2021) and Vasso (Li & Giannakis, 2024) with our loss landscape smoothing mechanism removed. For a fair comparison, the hyperparameters of each method are properly tuned for the adaption to all the tasks including Cifar100 with 1 IPC and Tiny ImageNet with 3 IPC. We repeat each method 5 times and report the mean and variance in Table 4. The results imply that all the sharpness methods consistently improve MTT (Cazenavette et al., 2022), which justifies the benefit of sharpness minimisation. However, the competitors all fail to defeat our method

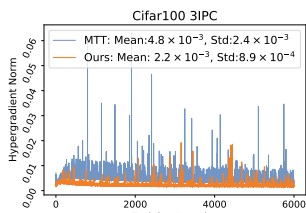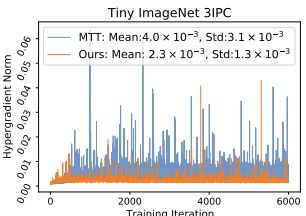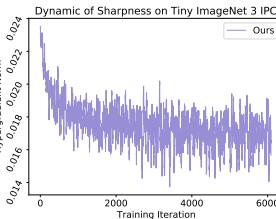

Figure 3: Sharpness analysis by visualisation. Hypergradient Norm comparison between MTT and SATM. Left: the hypergradient norm on Cifar100 with 10 IPC; Middle: the hypergradient norm on Tiny ImageNet with 3 IPC. Right: Sharpness dynamic on Tiny ImageNet with 3 IPC.

due to the failure to accurately compute the sharpness proxy. Moreover, EMA, equivalent to FTD without Sharpness-aware minimisers to generate expert trajectories, gains minimal improvement.

### 4.3.2 Hypergradient Analysis

To illustrate the effects of sharpness minimisation on the process of synthetic dataset learning, we record the hypergradient norm of MTT and SATM during training and report their mean and variance over training iterations. Depicted in Fig 3, SATM has a smaller mean and variance than MTT on Cifar100 with 3 IPC and Tiny ImnageNet 3IPC. Additionally, fewer spikes of hypergraident in SATM can be observed, indicating more stable training. Moreover, the dynamic of the sharpness, measured by $\mathcal{L}(\phi + \epsilon) - \mathcal{L}(\phi)$, with decreasing trend shows that the synthetic dataset is landing into the flat loss region.

### 4.3.3 Two Inner Loop Routine

Our method has a similar training protocol with TESLA (Cui et al., 2023), as both require executing the inner loop twice to enable outer loop updates. However, TESLA trades off time complexity in its two inner loops to maintain a constant memory cost that is agnostic to the unrolling inner loop steps. In contrast, our model also achieves constant memory usage by differentiating through the last N steps of the inner loop, thanks to provable hypergradient approximation error bound. Moreover, it requires only a partial second inner loop execution and aims to converge into a flat loss region improving the generalization of synthetic data significantly, outperforming TESLA even without relying on soft-label fitting tricks.

## 5 Conclusions, Limitations and Future Works

In this work, we explore the generalisation ability of condensed datasets produced by training trajectory-matching-based algorithms via jointly optimising the sharpness and the distance between real and synthetic trajectories. We propose Sharpness-Aware Trajectory Matching (SATM) to reduce the computational cost caused by the long horizon inner loop and the mini-max optimisation for the sharpness minimisation through the proposed hypergradient approximation strategies. Those strategies have clear theoretical motivation, limited error in practice, and a framework flexible enough to adapt to other sharpness-aware based algorithms. The improvement of the generalisation is observed in a variety of in- and out-of-domain tasks such as cross-architecture and cross-task (continual learning) with a comprehensive analysis of the algorithm's sharpness properties on the training dynamics.

Despite the superior performance of SATM, we observed that the proposed algorithm can potentially serve as a "plug-and-play" model for other dataset condensation methods and, more broadly, for various bilevel optimisation applications, such as loss function learning, optimiser learning and middle shot learning. However, these possibilities are not explored in this work and we leave them to the future work. Moreover, beyond focusing on reusing the trajectory to enhance training efficiency in reaching flat regions, future research could be in advanced gradient estimation directions, such as implicit gradients, showing promise for managing long-horizon inner loops and avoiding second-order unrolling. This could potentially eliminate the entire second trajectory resulting in higher computational efficiency and less approximation error.

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

## A  APPENDIX

### A.1  PROOF FOR THEOREM 3.2

**Theorem 3.2.** *Let $\mathcal{L}(\phi, \theta)$ be a function that is $\sigma$-smooth and continuous with respect to its arguments $\phi$ and $\theta$. Additionally, let the second-order derivatives $\nabla_\phi \nabla_\theta \mathcal{L}(\phi, \theta)$ be $\beta$-continuous. Consider two trajectories obtained by conducting gradient descent training on the datasets $\phi$ and $\phi + \epsilon$, respectively, with a carefully chosen learning rate $\alpha$ and identical initializations. After $\tau$ steps of training, let $\Delta\theta_\tau = \hat{\theta}_\tau(\phi + \epsilon) - \theta_\tau(\phi)$. Then, we have:*

$$\|\Delta\theta_\tau\| \leq \alpha\tau(2\sigma + \beta\rho).$$

*Proof.* Let:

$$\hat{\theta}_\tau = \theta_0 - \alpha \sum_i^\tau \nabla\mathcal{L}(\phi + \epsilon, \hat{\theta}_i)$$

$$\theta_\tau = \theta_0 - \alpha \sum_i^\tau \nabla\mathcal{L}(\phi, \theta_i)$$

then after N step iterations, the difference between $\theta_N$ and $\hat{\theta}_N$ is

$$\|\Delta\theta_\tau\| = \left\|\hat{\theta}_\tau - \theta_\tau\right\| = \left\|-\alpha\sum_i^\tau(\nabla\mathcal{L}(\phi + \epsilon, \hat{\theta}_i) - \nabla\mathcal{L}(\phi, \theta_i))\right\|$$

$$= \alpha\left\|\sum_i^\tau(\nabla\mathcal{L}(\phi + \epsilon, \hat{\theta}_i) - \nabla\mathcal{L}(\phi, \theta_i))\right\|$$

We compute the gradient difference:

$$||\nabla\mathcal{L}(\phi + \epsilon, \hat{\theta}_i) - \nabla\mathcal{L}(\phi, \theta_i)||$$
$$\approx ||\nabla\mathcal{L}(\phi, \hat{\theta}_i) + \nabla_\phi\nabla_\theta\mathcal{L}(\phi, \hat{\theta}_i) \cdot \epsilon - \nabla\mathcal{L}(\phi, \theta_i)||$$
$$\leq ||\nabla\mathcal{L}(\phi, \hat{\theta}_i) - \nabla\mathcal{L}(\phi, \theta_i)|| + ||\nabla_\phi\nabla_\theta\mathcal{L}(\phi, \hat{\theta}_i) \cdot \epsilon||$$
$$\leq 2\sigma + ||\nabla_\phi\nabla_\theta\mathcal{L}(\phi, \hat{\theta}_i)|| ||\epsilon||$$

With $\nabla_\phi\nabla_\theta\mathcal{L}(\phi, \hat{\theta}_i)$ is $\beta$ smooth and $||\epsilon|| = \rho$ :

$$||\nabla\mathcal{L}(\phi + \epsilon, \hat{\theta}_i) - \nabla\mathcal{L}(\phi, \theta_i)||_2 \leq 2\sigma + \beta\rho$$

Then:

$$\|\Delta\theta_\tau\| \leq \alpha\tau(2\sigma + \beta\rho)$$

$\square$

### A.2  PYTORCH BASED PSEUDOCODE FOR TRUNCATED UNROLLING HYPERGRADIENT

---

Algorithm 2: Trucated hypergradient computation

---
```
stop gradient:
```
**for** $i = 1, \ldots, \iota$ **do**
   $\theta_i = \theta_{i-1} - \alpha * \text{torch.grad}(\mathcal{L}_{CE}(\theta, \phi), \theta)$
**end for**
```
with gradient:
```
**for** $i = 1, \ldots, N - \iota$ **do**
   $\theta_i = \theta_{i-1} - \alpha * \text{torch.grad}(\mathcal{L}_{CE}(\theta, \phi), \theta, \text{retain\_graph} = \text{True}, \text{create\_graph} = \text{True})$
**end for**
**Return:** $\theta_N(\phi)$

---

## A.3 PROOF OF PROPOSITION 3.1

**Proposition 3.1.** *Assmue $\mathcal{L}_{CE}$ is $K$-smooth, twice differentiable, and locally $J$-strongly convex in $\theta$ around $\{\theta_{\iota+1}, ..., \theta_N\}$. Let $\Xi(\theta, \phi) = \theta - \alpha \nabla \mathcal{L}_{CE}(\theta, \phi)$. For $\alpha \leq \frac{1}{K}$, then*

$$\left\| \frac{\partial F(\phi)}{\partial \phi} - \frac{\partial F_\iota(\phi)}{\partial \phi} \right\| \leq 2^\iota (1 - \alpha J)^{N-\iota+1} \left\| \frac{\partial \mathcal{L}(\theta(\phi))}{\partial \theta_N(\phi)} \right\| \max_{i \in \{0,..\iota\}} \left\| \frac{\partial \theta_i}{\partial \phi} \right\|$$

*where $\frac{\partial F(\phi)}{\partial \phi}$ denotes the untruncated hypergradient.*

*Proof.* Let

$$A_{i+1} = \frac{\partial \theta_{i+1}}{\partial \theta_i}, B_{i+1} = \frac{\partial \theta_{i+1}}{\partial \phi}$$

then

$$\frac{\partial F(\phi)}{\partial \phi} = \frac{\partial \mathcal{L}(\theta(\phi))}{\partial \phi} + \sum_{i=0}^{N} B_i A_{i+1} \cdots A_N \frac{\partial \mathcal{L}(\theta(\phi))}{\partial \theta_N(\phi)}$$

Let $e_\iota = \frac{\partial F(\phi)}{\partial \phi} - \frac{\partial F_\iota(\phi)}{\partial \phi}$,

$$e_\iota = \left( \sum_{i=0}^{\iota} B_i A_{i+1} \cdots A_\iota \right) A_{\iota+1} \cdots A_N \frac{\partial \mathcal{L}(\theta(\phi))}{\partial \theta_N(\phi)}$$

Given $\mathcal{L}_{CE}$ is locally $J$-strongly convex with respect to $\theta$ in the neighborhood of $\{\theta_{\iota+1}, \ldots, \theta_N\}$,

$$\|e_\iota\| \leq \left\| \sum_{i=0}^{\iota} B_i A_{i+1} \cdots A_\iota \right\| \left\| A_{\iota+1} \cdots A_N \frac{\partial \mathcal{L}(\theta(\phi))}{\partial \theta_N(\phi)} \right\|$$

$$\leq (1 - \alpha J)^{N-\iota+1} \left\| \frac{\partial \mathcal{L}(\theta(\phi))}{\partial \theta_N(\phi)} \right\| \left\| \sum_{i=0}^{\iota} B_i A_{i+1} \cdots A_\iota \right\|$$

In the worst case, when $\mathcal{L}_{CE}$ is $K$-smooth but nonconvex, then if the smallest eigenvalue of $\frac{\partial^2 \mathcal{L}_{CE}(\theta, \phi)}{\partial \theta \, \partial \theta}$ is $-K$, then $\|A_i\| = 1 + \alpha K \leq 2$ for $i = 0, \ldots, \iota$. $\qquad \square$

## A.4 THE DERIVATION OF LEARNING RATE LEARNING WITH FIRST ORDER DERIVATIVE

In this section, we provide the derivation of the hypergradient calculation for learning rate $\alpha$. Given the outer loop objective, $\mathcal{L}(\theta(\phi))$, and the inner loop object $\mathcal{L}_{CE}(\theta_i, \phi)$ with $N$ iteration unrolling, the computation can be dedicated by:

$$\frac{\partial \mathcal{L}(\theta_N(\phi))}{\partial \alpha} = \frac{\partial \mathcal{L}(\theta_N(\phi))}{\partial \theta_N} \cdot \frac{\partial (\theta_N, \phi)}{\partial \alpha}$$

$$= \frac{\partial \mathcal{L}(\theta_N(\phi))}{\partial \theta_N} \cdot \frac{\partial \Xi(\theta_{N-1}, \phi)}{\partial \alpha}$$

$$= \frac{\partial \mathcal{L}(\theta_N(\phi))}{\partial \theta_N} \cdot \frac{\partial}{\partial \alpha} \left( \theta_{N-1} - \alpha \frac{\partial \mathcal{L}_{CE}(\theta_{N-1}, \phi)}{\partial \theta_{N-1}} \right)$$

$$= \frac{\partial \mathcal{L}(\theta_N(\phi))}{\partial \theta_N} \cdot \left( \frac{\partial \theta_{N-1}}{\partial \alpha} - \frac{\partial \mathcal{L}_{CE}(\theta_{N-1}, \phi)}{\partial \theta_{N-1}} \right)$$

$$\text{we treat } \frac{\partial \mathcal{L}_{CE}(\theta_{N-1}, \phi)}{\partial \theta_{N-1}} \text{ as a constant w.r.t. } \alpha$$

$$= \frac{\partial \mathcal{L}(\theta_N(\phi))}{\partial \theta_N} \cdot \left( \frac{\partial}{\partial \alpha} \Xi(\theta_{N-2}, \phi) - \frac{\partial \mathcal{L}_{CE}(\theta_{N-1}, \phi)}{\partial \theta_{N-1}} \right)$$

$$= \frac{\partial \mathcal{L}(\theta_N(\phi))}{\partial \theta_N} \cdot \left( - \sum_{i=0}^{N-1} \frac{\partial \mathcal{L}_{CE}(\theta_i, \phi)}{\partial \theta_i} \right)$$

## A.5 COMPUTATIONAL RESOURCE

We conduct all our experiments on two Tesla V100-32GB GPUs with Intel(R) Xeon(R) W-2245 CPU @ 3.90GHz and one A100-40GB GPU with Intel(R) Xeon(R) Gold 5118 CPU @ 2.30GHz which are on different servers. Thus, we cannot run the full batch of synthetic dataset learning as the same as other trajectory matching-based methods when the inner loop trajectories contain many unrolling iterations. Those cases include Cifar100-10IPC, Cifar100-50IPC, and Tiny ImageNet 1IPC. In our case, stochastic gradient descent with mini-batch is utilised in the outer loop instead.

## A.6 HYPERPARAMETERS AND EXPERIMENT DETAILS

The hyperparameters used for condensing datasets in all the settings are given in Tab 6 with ConvNet (Gidaris & Komodakis, 2018) applied to construct the training trajectories.

| Dataset | Model | IPC | Synthetic Steps ($N$) | Expert Epochs ($M$) | Max Start Epoch ($T$) | Synthetic Batch Size | ZCA | Learning Rate (Images) | Learning Rate (Step size) |
|---|---|---|---|---|---|---|---|---|---|
| CIFAR-10 | ConvNetD3 | 1 | 50 | 2 | 2 | - | Y | 1000 | $1\times10^{-6}$ |
| | | 3 | 50 | 2 | 2 | - | Y | 100 | $1\times10^{-5}$ |
| | | 10 | 30 | 2 | 20 | - | Y | 50 | $1\times10^{-5}$ |
| | | 50 | 30 | 2 | 40 | - | Y | 100 | $1\times10^{-5}$ |
| CIFAR-100 | ConvNetD3 | 1 | 40 | 3 | 20 | - | Y | 500 | $1\times10^{-5}$ |
| | | 3 | 45 | 3 | 20 | - | Y | 1000 | $5\times10^{-5}$ |
| | | 10 | 20 | 2 | 20 | 500 | Y | 1000 | $1\times10^{-5}$ |
| | | 50 | 80 | 2 | 40 | 500 | Y | 1000 | $1\times10^{-5}$ |
| Tiny ImageNet | ConvNetD4 | 1 | 30 | 2 | 10 | 200 | Y | 1000 | $1\times10^{-4}$ |
| | | 3 | 30 | 2 | 15 | 200 | Y | 1000 | $1\times10^{-4}$ |
| | | 10 | 20 | 2 | 40 | 200 | Y | 10000 | $1\times10^{-4}$ |

Table 6: Hyper-parameters used for our SATM. A synthetic batch size of "-" represents that a full batch set is used in each outer loop iteration. ConvNetD3 and ConvNet4D denote the 3-layer and 4-layer ConvNet (Gidaris & Komodakis, 2018) respectively. In all the settings, ZCA whitening (Nguyen et al., 2021b;a) is applied.

## A.7 COMPUTATIONAL COST COMPARSION

We computed and recorded the memory and time costs when running SATM and then compared them with MTT and TESLA following Tesla's experimental protocol. The results were primarily measured on a single NVIDIA A6000 GPU, except for MTT on ImageNet-1K (Russakovsky et al., 2015), which required two A6000 GPUs.

In most of our experiments, only one-third of the inner loop is retained to compute the hypergradients for sharpness approximation and synthetic dataset optimization. In the worst-case scenario, we keep half of the inner loop to ensure training stability and efficiency. Given the result in Table 7, our strategy significantly reduces memory consumption compared to MTT, enabling the dataset to be trained on a single A6000 GPU.

| | MTT Memory | TESLA Memory | SATM (N/2) Memory | SATM (N/3) Memory |
|---|---|---|---|---|
| CIFAR-100 | 17.1±0.1 GB | 3.6±0.1 GB | 8.7±0.1 GB | 5.7±0.1 GB |
| ImageNet-1K | 79.9±0.1 GB | 13.9±0.1 GB | 39.6±0.1 GB | 26.6±0.1 GB |

Table 7: Comparison of memory usage across different methods and datasets. We refer to the cases where one-third and one-half of the inner loop are retained as SATM (N/3) and SATM (N/2), respectively.

In terms of time cost illustrated in Table 8, SATM consistently outperforms the two inner-loop-based algorithms, Tesla. In the one-third inner loop case, SATM even consumes less time than MTT which requires retaining a full single inner loop.

|           | MTT Time       | TESLA Time     | SATM (N/2) Time | SATM (N/3) Time |
|-----------|----------------|----------------|-----------------|-----------------|
| CIFAR-100 | 12.1±0.6 sec   | 15.3±0.5 sec   | 12.8±0.6 sec    | 12.0±0.5 sec    |
| ImageNet-1K | 45.9±0.5 sec | 47.4±0.7 sec   | 46.1±0.4 sec    | 45.4±0.4 sec    |

Table 8: Comparison of execution time across different methods and datasets. We refer to the cases where one-third and one-half of the inner loop are retained as SATM (N/3) and SATM (N/2), respectively.

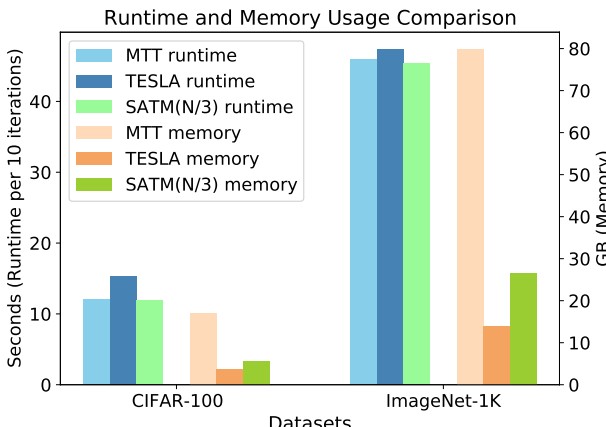

Figure 4: GPU memory and runtime comparison among MTT, TESLA and SATM (N/3) on CIFAR100 and ImageNet-1K with results measured with a batch size of 100 and 50 inner loop steps.

To further justify the memory efficiency of SATM, we challenge the ImageNet-1K setting following the training and evaluation protocol from Tesla. By truncating the inner loop computational graph hold for hypergradient computation, SATM is executable on the heavy memory setting with results given in Table 9.

| Dataset | IPC | TESLA | SATM |
|---------|-----|-------|------|
| ImageNet-1K | 1 | 7.7±0.2 | **8.2**±0.4 |
|  | 2 | 10.5±0.3 | **11.4**±0.2 |
|  | 10 | 17.8±1.3 | **18.5**±0.9 |
|  | 50 | 27.9±1.2 | **28.4**±1.1 |

Table 9: Comparison of TESLA and SATM across different IPCs on ImageNet-1K.

## A.8   FLAT INNER LOOP STUDY

SATM is developed based on MTT without incorporating the components introduced in FTD (Du et al., 2023), particularly the expert trajectories generated by sharpness-aware optimizers such as GSAM. However, understanding whether SATM can be compatible with advanced expert trajectories is desirable to study. Therefore, we follow the expert trajectory generation protocol and execute SATM on the flat expert trajectories with the results in Table 10. It can be observed that the inclusion of a flat inner loop leads to clear improvements in SATM-FI compared to both standard SATM and FTD. Furthermore, the authors of FTD noted the limited performance contribution of EMA, which was originally intended to guide the synthetic dataset toward convergence on a flat loss landscape. SATM addresses this limitation and effectively demonstrates the benefits of leveraging flatness for improved generalization.

| | IPC | MTT | FTD | SATM | SATM-FI |
|---|---|---|---|---|---|
| CIFAR-10 | 1 | 46.2±0.8 | 46.8±0.3 | 49.0±0.3 | **48.7**±0.4 |
| | 10 | 65.4±0.7 | 66.6±0.3 | 67.1±0.4 | **67.9**±0.3 |
| | 50 | 71.6±0.2 | 73.8±0.2 | 73.9±0.2 | **74.2**±0.4 |
| CIFAR-100 | 1 | 24.3±0.3 | 25.2±0.2 | 26.1±0.4 | **26.6**±0.5 |
| | 10 | 39.7±0.4 | 43.4±0.3 | 43.1±0.5 | **43.9**±0.7 |
| | 50 | 47.7±0.2 | 50.7±0.3 | 50.9±0.5 | **51.4**±0.5 |
| Tiny-ImageNet | 1 | 8.8±0.3 | 10.4±0.3 | 10.9±0.2 | **11.7**±0.4 |
| | 10 | 23.2±0.1 | 24.5±0.2 | 25.4±0.4 | **25.6**±0.6 |

Table 10: Accuracy (%) Comparison of MTT, FTD, SATM, and SATM-FI across different datasets and configurations.

## A.9 Truncated Step Study

We chose the settings that require the long inner loops for dataset learning to study the correlation between the number of inner loop steps remaining for differentiation and the model performance. Table 11 details the experimental settings, including the dataset, the number of images per category (IPC), and the inner loop steps $N$. For example, "CIFAR-10 (1 IPC, 50 steps)" refers to condensing one synthetic image per category with 50 inner loop steps. To analyze the effect on performance, we retained the last $\frac{1}{k}$ steps, where $k = 2, 3, 4, 5, 6$, of the total inner loop steps. For simplicity, the inner loop steps remained for the first round of hypergradient computation and trajectory reusing in the second round is kept the same which is applied across all experiments. The operation $int(\frac{N}{k})$ is used to determine the remaining inner loop steps. We examined how accuracy changes with the remaining inner loop steps by executing SATM for 10000 training iterations. A clear trend emerged: performance improves as the number of truncated iterations decreases and converges once the differentiation steps reach a certain threshold.

| Configuration/Steps | $\frac{1}{6}$ | $\frac{1}{5}$ | $\frac{1}{4}$ | $\frac{1}{3}$ | $\frac{1}{2}$ |
|---|---|---|---|---|---|
| CIFAR-10 (1IPC, 50step) | 45.2 | 48.8 | 47.5 | 49.0 | 49.2 |
| CIFAR-100 (50IPC, 80step) | 23.4 | 33.4 | 48.7 | 50.9 | 50.5 |

Table 11: Accuracy (%) change along with the truncated inner loop step change on CIFAR-10 and CIFAR-100 datasets.

## A.10 More related work and comparison with Recent Method

A recent method, RDED (Sun et al., 2024), introduces new perspectives to the dataset distillation field by constructing synthetic images from original image crops and labelling them with a pre-trained model. In comparison, our work falls within the training trajectory matching area and focuses on efficient bilevel optimization with a long inner loop with the goal of enhancing the generalization ability of synthetic data by developing an efficient, sharpness-aware optimizer for bilevel optimization.

DATM (Guo et al., 2024) utilizes the difficulty of training trajectories to implement a curriculum learning-based dataset condensation protocol. While this approach is relevant, it is somewhat distinct from research focused on optimization efficiency and generalization, such as Tesla, FTD, and SATM, which prioritize optimization efficiency through gradient approximation. Additionally, from an implementation perspective, DATM feeds expert trajectories in an easy-to-hard sequence directly into FTD. In contrast, our work focuses on the flatness of the loss landscape of the learning dataset from a bilevel optimization perspective, rather than emphasizing pure performance comparisons. Nevertheless, we believe our method is compatible with DATM. To demonstrate this, we conducted experiments combining DATM's easy-to-hard training protocol with SATM, yielding the following results in Table 12.

|  | IPC | MTT | FTD | DATM | SATM-DA |
|---|---|---|---|---|---|
| CIFAR-10 | 1 | $46.2 \pm 0.8$ | $46.8 \pm 0.3$ | $46.9 \pm 0.5$ | $\mathbf{48.6 \pm 0.4}$ |
|  | 10 | $65.4 \pm 0.7$ | $66.6 \pm 0.3$ | $66.8 \pm 0.2$ | $\mathbf{68.1 \pm 0.3}$ |
|  | 50 | $71.6 \pm 0.2$ | $73.8 \pm 0.2$ | $76.1 \pm 0.3$ | $\mathbf{76.4 \pm 0.6}$ |
| CIFAR-100 | 1 | $24.3 \pm 0.3$ | $25.2 \pm 0.2$ | $27.9 \pm 0.2$ | $\mathbf{28.2 \pm 0.8}$ |
|  | 10 | $39.7 \pm 0.4$ | $43.4 \pm 0.3$ | $47.2 \pm 0.4$ | $\mathbf{48.3 \pm 0.4}$ |
|  | 50 | $47.7 \pm 0.2$ | $50.7 \pm 0.3$ | $55.0 \pm 0.2$ | $\mathbf{55.7 \pm 0.3}$ |
| Tiny-ImageNet | 1 | $8.8 \pm 0.3$ | $10.4 \pm 0.3$ | $\mathbf{17.1 \pm 0.3}$ | $16.4 \pm 0.4$ |
|  | 10 | $23.2 \pm 0.1$ | $24.5 \pm 0.2$ | $31.1 \pm 0.3$ | $\mathbf{32.3 \pm 0.6}$ |

Table 12: Accuracy (%) Comparison of MTT, FTD, DATM, and SATM-DA across different IPCs, datasets and configurations.

### A.11   ILLUSTRATION FOR THE SYNTHETIC IMAGES

We visualise the learned synthetic datasets on Cifar10, Cifar100 and Tiny ImageNet in this section.

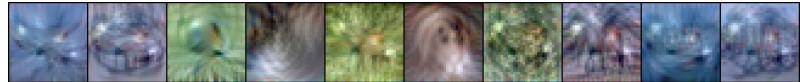

Figure 5: Cifar10 with 1IPC

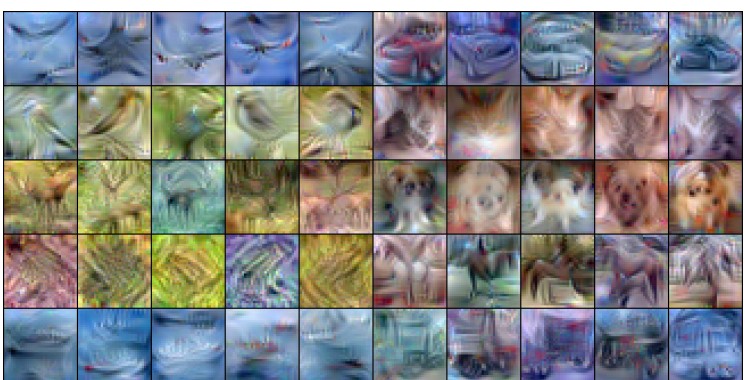

Figure 6: Cifar10 with 3IPC

Figure 7: Cifar10 with 10IPC

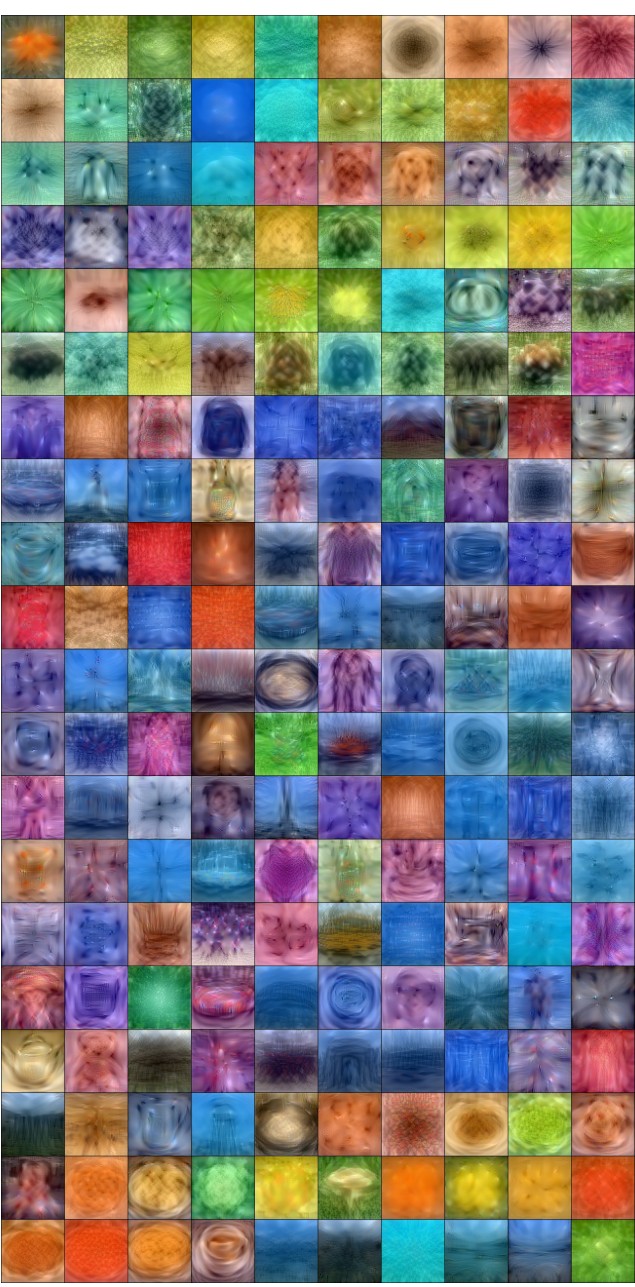

Figure 8: Cifar100 with 1IPC

