# OpenReview forum: "Dataset Condensation with Sharpness-Aware Trajectory Matching"
_ICLR.cc/2025/Conference — Submitted to ICLR 2025_

### Official Review · Reviewer_MAmu · 2024-10-30

**Soundness:** 3
**Presentation:** 3
**Contribution:** 2
**Rating:** 6
**Confidence:** 4

**Summary:**

This paper proposes to improve the performance of trajectory-matching based dataset distillation method by using sharpness-aware minimiser (SAM) to flatten the trajectory used for matching.
Also, to mitigate the additional budget introduced by SAM, several tricks including truncate unrolling hypergradient and trajectory resuing are proposed.

**Strengths:**

1. Good writing, easy to follow.
2. The author gives comprehensive math proof of how to efficiently integrate SAM into trajectory matching-based distillation method, improving this paper's soundness.

**Weaknesses:**

1. Missing comparison with SOTA methods such as DATM [1] in Table 2.

2. Only one ablation study (on sharpness-aware optimization method) is conducted. Authors should also conduct ablation studies on the proposed trajectory resuing and hyper-gradient truncating methods to prove their effectiveness.

3. Poor performance. Compared with FTD [2], which also proposes to use SAM, the performance improvement brought by SATM becomes marginal as IPC increases (less than 0.2% on CIFAR, IPC50). Is this method only effective in low IPC cases?

4. The distillation cost is only calculated in theory. Comparisons of the distillation cost in practice should be included.

5. The evaluation setting is not introduced clearly, whether the zca whitening, EMA, and DSA augment are used?

[1] Towards lossless dataset distillation via difficulty-aligned trajectory matching, ICLR 2024.

[2] Minimizing the Accumulated Trajectory Error to Improve Dataset Distillation, CVPR 2023.

**Questions:**

Please see weakness

---

> ### Author Response · Authors · 2024-11-21
> **Authers' response**
>
> ###  ***Weakness 1: missing comparison with DATM***
> We thank the reviewers for highlighting this point. DATM [1] utilizes the difficulty of training trajectories to implement a curriculum learning-based dataset condensation protocol. While this approach is relevant, it is somewhat distinct from research focused on optimization efficiency and generalization, such as Tesla [2], FTD [3], and SATM, which prioritize optimization efficiency through gradient approximation. Additionally, from an implementation perspective, DATM feeds expert trajectories in an easy-to-hard sequence directly into FTD. In contrast, our work focuses on the flatness of the loss landscape of the learning dataset from a bilevel optimization perspective, rather than emphasizing pure performance comparisons. Nevertheless, we believe our method is compatible with DATM. To demonstrate this, we conducted experiments combining DATM's easy-to-hard training protocol with SATM, yielding the following results:
>
> ***Accuracy (%) comparsion with DATM***
>
>
> |                 |  IPC   | MTT          |     FTD          |    DATM          |    SATM-DA           |
> |-----------------|--------|--------------|------------------|------------------|----------------------|
> |                 |  1     | 46.2$\pm$0.8 |  46.8$\pm$0.3    |  46.9$\pm$0.5    | **48.6**$\pm$0.4     |
> | CIFAR-10        |  10    | 65.4$\pm$0.7 |  66.6$\pm$0.3    |  66.8$\pm$0.2    | **68.1**$\pm$0.3     |
> |                 |  50    | 71.6$\pm$0.2 |  73.8$\pm$0.2    |  76.1$\pm$0.3    | **76.4**$\pm$0.6     |
> |                 |  1     | 24.3$\pm$0.3 |  25.2$\pm$0.2    |  27.9$\pm$0.2    | **28.2**$\pm$0.8     |
> | CIFAR-100       |  10    | 39.7$\pm$0.4 |  43.4$\pm$0.3    |  47.2$\pm$0.4    | **48.3**$\pm$0.4     |
> |                 |  50    | 47.7$\pm$0.2 |  50.7$\pm$0.3    |   55.0$\pm$0.2   | **55.7**$\pm$0.3     |
> | Tiny-ImageNet   |  1     |  8.8$\pm$0.3 |  10.4$\pm$0.3    |  **17.1**$\pm$0.3    | 16.4$\pm$0.4     |
> |                 |  10    | 23.2$\pm$0.1 |  24.5$\pm$0.2    |  31.1$\pm$0.3    | **32.3**$\pm$0.6     |
>
> Thanks to the straightforward difficulty alignment proposed in DATM, integrating SATM into DATM (resulting in SATM-DA) allows SATM-DA to benefit from both the curriculum learning strategy and the flatness, leading to superior performance compared to DATM. We will discuss this phenomenon involving DATM and SATM-DA in our paper.
>
> ###  ***Weakness 2: additional ablation study***
> We agree with the reviewer that additional ablation studies would strengthen our submission. In addition to the ablation study on the learning rate approximation and the flatness introduced by different optimizers, we have followed the reviewer’s suggestion to investigate the impact of the number of truncated inner loop iterations on performance.
>
> We chose the settings that require long inner loops for dataset learning. The table below details the experimental settings, including the dataset, the number of images per category (IPC), and the inner loop steps $N$. For example, "CIFAR-10 (1 IPC, 50 steps)" refers to condensing one synthetic image per category with 50 inner loop steps. To analyze the effect on performance, we retained the last $\frac{1}{k}$ steps, where $k =2,3,4,5,6$, of the total inner loop steps. For simplicity, the inner loop steps remained for the first round of hypergradient computation and trajectory reusing in the second round is kept the same which is applied across all experiments. The operation $int(\frac{N}{k})$ is used to determine the remaining inner loop steps. We examined how accuracy changes with the remaining inner loop steps by executing SATM for 10000 training iterations. A clear trend emerged: performance improves as the number of truncated iterations decreases and converges once the differentiation steps reach a certain threshold.
>
> ***Accuracy (%) change with the remaining inner loop step***
>
> |   setting\steps             |  1/6    |  1/5  | 1/4   | 1/3  | 1/2   |
> |-----------------------------|---------|-------|-------|------|-------|
> | CIFAR-10  (1IPC,  50step)   |  45.2   |  48.8 | 47.5  | 49.0 | 49.2  |
> | CIFAR-100 (50IPC, 80step)   |  23.4   |  33.4 | 48.7  | 50.9 | 50.5  |
>
> ----------- to continue -------------

---

> ### Author Response · Authors · 2024-11-21
> **Authers' response**
>
> ###  ***Weakness 3: comparison with FTD***
> SATM is developed based on MTT without incorporating the components introduced in FTD, particularly the expert trajectories generated by sharpness-aware optimizers such as GSAM. Therefore, a direct comparison between SATM and FTD may not provide a fair assessment of whether SATM can generalize to high IPC settings. However, when compared to MTT under high IPC conditions, SATM demonstrates a clear improvement margin, supporting its ability to generalize effectively in such scenarios.
>
> To address the reviewer's questions and concerns regarding the performance of our algorithm, we additionally evaluate SATM using expert trajectories produced by sharpness-aware optimizers. This variant, referred to as SATM-FI, is discussed, and the results are presented below.
>
> |                 |  IPC   |  MTT         |     FTD          |      SATM        | SATM-FI          |
> |-----------------|--------|--------------|------------------|------------------|------------------|
> |                 |  1     | 46.2$\pm$0.8 |  46.8$\pm$0.3    |  49.0$\pm$0.3    | **48.7**$\pm$0.4 |
> | CIFAR-10        |  10    | 65.4$\pm$0.7 |  66.6$\pm$0.3    |  67.1$\pm$0.4    | **67.9**$\pm$0.3     |
> |                 |  50    | 71.6$\pm$0.2 |  73.8$\pm$0.2    |  73.9$\pm$0.2    | **74.2**$\pm$0.4     |
> |                 |  1     | 24.3$\pm$0.3 |  25.2$\pm$0.2    |  26.1$\pm$0.4    | **26.6**$\pm$0.5     |
> | CIFAR-100       |  10    | 39.7$\pm$0.4 |  43.4$\pm$0.3    |  43.1$\pm$0.5    | **43.9**$\pm$0.7     |
> |                 |  50    | 47.7$\pm$0.2 |  50.7$\pm$0.3    |  50.9$\pm$0.5    | **51.4**$\pm$0.5     |
> | Tiny-ImageNet   |  1     |  8.8$\pm$0.3 |  10.4$\pm$0.3    |  10.9$\pm$0.2    | **11.7**$\pm$0.4     |
> |                 |  10    | 23.2$\pm$0.1 |  24.5$\pm$0.2    |  25.4$\pm$0.4    | **25.6**$\pm$0.6     |
>
> It can be observed that the inclusion of a flat inner loop leads to clear improvements in SATM-FI compared to both standard SATM and FTD. Furthermore, the authors of FTD noted the limited performance contribution of EMA, which was originally intended to guide the synthetic dataset toward convergence on a flat loss landscape. SATM addresses this limitation and effectively demonstrates the benefits of leveraging flatness for improved generalization.
>
> ###  ***Weakness 4: empirical memory and time cost***
> We computed and recorded the memory and time costs when running SATM and then compared them with MTT and Tesla following Tesla's experimental protocol. The results were primarily measured on a single NVIDIA A6000 GPU, except for MTT on ImageNet-1K, which required two A6000 GPUs.
>
> In most of our experiments, only one-third of the inner loop is retained to compute the hypergradients for sharpness approximation and synthetic dataset optimization. In the worst-case scenario, we keep half of the inner loop to ensure training stability and efficiency. In both cases, our strategy significantly reduces memory consumption compared to MTT, enabling the dataset to be trained on a single GPU. We refer to the cases where one-third and one-half of the inner loop are retained as SATM (N/3) and SATM (N/2), respectively.
>
> In terms of time cost, SATM consistently outperforms the two inner-loop-based algorithms, Tesla. In the 1/3 inner loop case, SATM even consumes less time than MTT which requires retaining a full single inner loop.
>
> |             |      MTT Memory   | Tesla Memory   | SATM (N/2) Memory | SATM (N/3) Memory|
> |-------------|-------------------|----------------|------------------|------------------|
> | CIFAR-100   |  17.1$\pm$0.1 GB |  3.6$\pm$0.1 GB |  8.7$\pm$0.1 GB |  5.7$\pm$0.1 GB  |
> | ImageNet-1K |  79.9$\pm$0.1 GB | 13.9$\pm$0.1 GB | 39.6$\pm$0.1 GB | 26.6$\pm$0.1 GB  |
>
> |             |      MTT Time      | Tesla Time      | SATM (N/2) Time   | SATM (N/3) Time   |
> |-------------|--------------------|-----------------|-------------------|-------------------|
> | CIFAR-100   |  12.1$\pm$0.6 sec | 15.3$\pm$0.5 sec | 12.8$\pm$0.6 sec | 12.0$\pm$0.5 sec  |
> | ImageNet-1K |  45.9$\pm$0.5 sec | 47.4$\pm$0.7 sec | 46.1$\pm$0.4 sec | 45.4$\pm$0.4 sec  |
>
> ----------- to continue -------------

---

> ### Author Response · Authors · 2024-11-24
>
> ###  ***Weakness 5: experiment settings***
> We apologize for any confusion regarding the experiment settings. We followed the MTT protocol, applying ZCA whitening and DSA augmentation, which includes colour jittering, cropping, cutout, flipping, scaling, and rotation, in all our experiments. We would like to emphasize that EMA is not applied in SATM. Since EMA also introduces flatness, it would be difficult to distinguish whether any improvements come from EMA itself. Additionally, the authors of FTD noted that EMA does not consistently contribute to performance improvements.
>
> ***Reference***
>
> [1] Guo Z, Wang K, Cazenavette G, LI H, Zhang K, You Y. Towards Lossless Dataset Distillation via Difficulty-Aligned Trajectory Matching. In The Twelfth International Conference on Learning Representations, 2024.
>
> [2] Cui J, Wang R, Si S, Hsieh CJ. Scaling up dataset distillation to imagenet-1k with constant memory. In International Conference on Machine Learning, 2023.
>
> [3] Du, J., Jiang, Y., Tan, V.Y., Zhou, J.T. and Li, H. Minimizing the accumulated trajectory error to improve dataset distillation. In Proceedings of the IEEE/CVF conference on computer vision and pattern recognition, 2023.

---

> ### Comment · Reviewer_MAmu · 2024-11-25
>
> The authors' response has addressed most of my concerns, so raise my score to 6.

---

> > ### Author Response · Authors · 2024-11-25
> >
> > We greatly appreciate the time and effort the reviewer dedicated to evaluating our submission, as well as the positive recognition reflected in the revised score. We are open to further discussions and are committed to making additional improvements to our work.

---

### Official Review · Reviewer_yFpF · 2024-11-02

**Soundness:** 2
**Presentation:** 2
**Contribution:** 2
**Rating:** 5
**Confidence:** 5

**Summary:**

The paper introduces a new approach of the Dataset Distillation method with Sharpness-Aware Minimisation (SAM).

**Strengths:**

The theoretical analysis and proofs are sufficient. The authors detailed the approach of Matching Training Trajectory (MTT) with SAM. The proof of the theorem is provided and technically sound.

The manuscript is well-structured.

**Weaknesses:**

Though the proposed approach is technically sound, my main concern is that the experiments can’t support the claims of the proposed model. Specifically,

1. The proposed method claims to reduce the computation overhead and the time complexity of the MTT. However, the method doesn’t provide a computational cost comparison with the current methods.

2. The proposed method claims that the memory cost is also reduced. Given that the baseline Tesla [1] mentioned in the manuscript already reduces the memory cost so that the distillation of ImageNet-1K becomes available, evaluating the effectiveness of the proposed method on ImageNet would be beneficial.

[1] Justin Cui, Ruochen Wang, Si Si, Cho-Jui Hsieh. “Scaling Up Dataset Distillation to ImageNet-1K with Constant Memory,” NeurIPS 2022.

**Questions:**

see weaknesses

---

> ### Author Response · Authors · 2024-11-21
> **Authers' response**
>
> We appreciate the reviewer’s positive comments on the quality of our mathematical analysis and writing. Below, we address the concerns raised.
>
> ###  ***Weakness 1: empirical memory and time cost***
> We thank the reviewer for pointing out this issue. In response, we conducted an empirical analysis of the time and memory space costs, with the results provided in the table below. We computed and recorded the memory and time costs when running SATM, and then compared them with MTT [1] and Tesla [2] following the experimental protocol from Tesla. The results were primarily measured on a single NVIDIA A6000 GPU, except for MTT on ImageNet-1k, which required two A6000 GPUs.
>
> In most of our experiments, only one-third of the inner loop is retained to compute the hypergradients for sharpness approximation and synthetic dataset optimization. In the worst-case scenario, we keep half of the inner loop to ensure training stability and efficiency. In both cases, our strategy significantly reduces memory consumption compared to MTT, enabling the dataset to be trained on a single GPU. We refer to the cases where one-third and one-half of the inner loop are retained as SATM (N/3) and SATM (N/2), respectively.
>
> In terms of time cost, SATM consistently outperforms the two inner-loop-based algorithms, Tesla. In the 1/3 inner loop case, SATM even consumes less time than MTT which requires retaining a full single inner loop.
>
> |             |      MTT Memory   | Tesla Memory   | SATM (N/2) Memory | SATM (N/3) Memory|
> |-------------|-------------------|----------------|------------------|------------------|
> | CIFAR-100   |  17.1$\pm$0.1 GB |  3.6$\pm$0.1 GB |  8.7$\pm$0.1 GB |  5.7$\pm$0.1 GB  |
> | ImageNet-1K |  79.9$\pm$0.1 GB | 13.9$\pm$0.1 GB | 39.6$\pm$0.1 GB | 26.6$\pm$0.1 GB  |
>
> |             |      MTT Time      | Tesla Time      | SATM (N/2) Time   | SATM (N/3) Time   |
> |-------------|--------------------|-----------------|-------------------|-------------------|
> | CIFAR-100   |  12.1$\pm$0.6 sec | 15.3$\pm$0.5 sec | 12.8$\pm$0.6 sec | 12.0$\pm$0.5 sec  |
> | ImageNet-1K |  45.9$\pm$0.5 sec | 47.4$\pm$0.7 sec | 46.1$\pm$0.4 sec | 45.4$\pm$0.4 sec  |
>
>
> The results above are updated in ***Appendix A.7*** in the latest version.
>
> We further study the relationship between the number of truncated inner loop steps and the synthetic datasets performance to proof the memory efficiency of SATM. We chose the settings that require long inner loops for dataset learning. The table below details the experimental settings, including the dataset, the number of images per category (IPC), and the inner loop steps $N$. For example, "CIFAR-10 (1 IPC, 50 steps)" refers to condensing one synthetic image per category with 50 inner loop steps. To analyze the effect on performance, we retained the last $\frac{1}{k}$ steps, where $k =2,3,4,5,6$, of the total inner loop steps. For simplicity, the inner loop steps remained for the first round of hypergradient computation and trajectory reusing in the second round is kept the same which is applied across all experiments. The operation $int(\frac{N}{k})$ is used to determine the remaining inner loop steps. We examined how accuracy changes with the remaining inner loop steps by executing SATM for a fixed number of training iterations (10,000). A clear trend emerged: performance improves as the number of truncated iterations decreases and converges once the differentiation steps reach a certain threshold.
>
> ***Accuracy (%) change with the remaining inner loop step***
>
> |   setting\steps             |  1/6    |  1/5  | 1/4   | 1/3  | 1/2   |
> |-----------------------------|---------|-------|-------|------|-------|
> | CIFAR-10  (1IPC,  50step)   |  45.2   |  48.8 | 47.5  | 49.0 | 49.2  |
> | CIFAR-100 (50IPC, 80step)   |  23.4   |  33.4 | 48.7  | 50.9 | 50.5  |
>
> ----------- to continue -------------

---

> ### Author Response · Authors · 2024-11-26
>
> ###  ***Weakness 2: experiments on ImageNet 1K***
> SATM undoubtedly requires less memory compared to MTT, as demonstrated in the previous section. We agree with the reviewer that testing SATM in a scaled-up setting is an excellent suggestion. Following Tesla's protocol, we evaluate SATM on ImageNet 1K. Additionally, we apply soft labels used in Tesla to SATM to ensure a fair comparison.
>
> |    Dataset    | IPC |    Tesla          |        SATM             |
> |---------------|-----|-------------------|-------------------------|
> |  ImageNet-1K  |  1  |   7.7$\pm$0.2     |    **8.2**$\pm$0.4      |
> |               |  2  |  10.5$\pm$0.3     |    **11.4**$\pm$0.2     |
> |               | 10  |  17.8$\pm$1.3     |    **18.5**$\pm$0.9     |
> |               | 50  |  27.9$\pm$1.2     |    **28.4**$\pm$1.1     |
>
> ***Reference***
>
> [1] Cui J, Wang R, Si S, Hsieh CJ. Scaling up dataset distillation to imagenet-1k with constant memory. In International Conference on Machine Learning, 2023.
>
> [2] Du, J., Jiang, Y., Tan, V.Y., Zhou, J.T. and Li, H. Minimizing the accumulated trajectory error to improve dataset distillation. In Proceedings of the IEEE/CVF conference on computer vision and pattern recognition, 2023.

---

> ### Author Response · Authors · 2024-11-26
>
> Dear reviewer,
>
> Thank you for sharing your concerns regarding our submission. With the extended rebuttal period, we are confident there is sufficient time to address your comments comprehensively. We have conducted the experiment as per your requirements to substantiate our method. We kindly request your feedback on our rebuttal at your earliest convenience.
>
> Best regards,
> Authors

---

> > ### Comment · Reviewer_yFpF · 2024-12-02
> >
> > Thank you for the rebuttal. However, I do not believe my concerns about the effectiveness on ImageNet has been adequately addressed. Tesla is not a strong baseline on ImageNet. Even comparing with Tesla, the improvement is not clear. Therefore, I will maintain my score.

---

> > > ### Author Response · Authors · 2024-12-03
> > >
> > > Thank you for your response. Given the limited rebuttal period, we were only able to complete the computationally expensive ImageNet-1K experiments with the model not fully tuned, which may explain the modest improvement. In our defence, we followed the reviewer’s suggestion to include a comparison with Tesla and conducted our experiments following its settings. Denying this baseline at the last minute seems unfair.
> > > Additionally, when fair hyperparameter tuning is applied, our method demonstrates a clear performance margin over Tesla in other settings. Besides, we believe we have effectively addressed the memory-related concerns highlighted in the first weakness by providing a detailed comparison of time and memory costs observed in practice.

---

### Official Review · Reviewer_JEkB · 2024-11-08

**Soundness:** 3
**Presentation:** 3
**Contribution:** 3
**Rating:** 5
**Confidence:** 2

**Summary:**

This article mainly builds on the MTT method, which aims to make a network mimic the training patterns of real data by ensuring that the paths (or parameter trajectories) created with synthetic data align with those from real data. The improvement here is adding a "sharpness-smoothing" strategy with Bayesian Optimization (BO) to approximate sharpness, plus a few other gradient tweaks.

**Strengths:**

This article does a solid job of exploring optimization theory, with most of the formula derivations being accurate. The experimental setup is also fairly thorough.

**Weaknesses:**

This article seems like a patchwork of various innovative points. The main concern is that the baseline experimental data is identical to that in the reference paper, Minimizing the Accumulated Trajectory Error to Improve Dataset Distillation. While it's understandable for the data to be similar, it raises the question: was the author involved in the previous paper? If so, please clarify this connection.

**Questions:**

1. The innovation in the method isnt major. It mostly combines BO with MTT. There's no new data compression method, just some changes to the optimization steps.
2. For improving data compression, the focus should be on synthesizing effective datasets to cut down the need for lots of data, not on cutting the cost of loop optimization.
3.Also, for experiments with DC, DM, MMT, etc., the results are the same as those in the paper "Minimizing the Accumulated Trajectory Error to Improve Dataset Distillation." Please list specific settings like the learning rate.

---

> ### Author Response · Authors · 2024-11-21
> **Authers' Response**
>
> ###  ***Weakness 1:***
> We agree with the reviewer that our work extends MTT [1] and studies in the line of research about trajectory matching-based dataset condensation, such as DATM [2], Tesla [3], and FTD [4]. We apologize if the way we presented our algorithm made its components appear somewhat disjointed. Here, we rephrase our motivation and logic to better link these algorithm components.
>
> We focus on the flatness of the loss landscape in dataset condensation, particularly when a long inner loop horizon is needed for optimizing the main free parameters—synthetic datasets. This type of optimization problem is challenged by a complicated outer loop loss landscape due to an unstable inner loop and the hypergradient issue. As a result, the learning process can converge to suboptimal points, degrading the performance of the produced datasets. To address this, we design SATM with the following main contributions:
>
> - We incorporate flatness into the outer loss landscape aiming to the generalisation ability improvement.
>
> - We tackle the computational overhead associated with the sharpness approximation and inner loop unrolling while also mitigating hypergradient problems such as gradient vanishing and explosion.
>
> - We propose constructing gradients with smooth ascent. This approach ensures stable sharpness and partially approximates the second inner loop unrolling by leveraging historical information from the first inner loop unrolling.
>
> In addition, the approximation errors introduced by the latter two are analyzed mathematically through ***Proposition 3.1*** and ***Theorem 3.2***.
>
> Regarding the experimental results, we cite the results from the FTD paper for DC [5], DM [6] and MTT [1], as they use the exact same setting. We clarify this in our paper. We also run DC, DM and MTT with their open-source codes.
>
> | Dataset            | IPC    | DC                      |    DC (Ours)                | DM                           | DM (ours)                |  MTT                      |  MTT  (ours)                       | FTD                        | TESLA                      | Ours                                     |
> |----------------|------|-----------------|----------------------|--------------------|--------------------|------------------|---------------------|-------------------|--------------------|---------------------------|
> |  Cifar-10          | 1        | $28.3_{\pm0.5}$ | $28.5_{\pm0.3}$      | $26.0_{\pm0.8}$    |  $26.7_{\pm0.5}$     |  $46.2_{\pm0.8}$  |  $46.7_{\pm0.6}$   | $46.8_{\pm0.3}$   | $48.5_{\pm0.8}$    |  $\textbf{49.0}_{\pm0.3}$  |
> |                          | 10     | $44.9_{\pm0.5}$  |$44.7_{\pm0.6}$       | $48.9_{\pm0.6}$    | $49.1_{\pm0.4}$      | $65.4_{\pm0.7}$   | $64.7_{\pm0.8}$   | $66.6_{\pm0.3}$    | $66.4_{\pm0.8}$   | $\textbf{67.1}_{\pm0.3}$ |
> |                          | 50     | $53.9_{\pm0.3}$  | $53.4_{\pm0.5}$      | $63.0_{\pm0.4}$     | $63.2_{\pm0.6}$    | $71.6_{\pm0.2}$   | $70.3_{\pm0.8}$   | $73.8_{\pm0.3}$    | $72.6_{\pm0.7}$    |  $\textbf{73.9}_{\pm0.2}$ |
> |  Cifar-100        | 1        | $12.8_{\pm 0.3}$ |   $12.4_{\pm0.5}$    | $11.4_{\pm0.3}$    | $10.9_{\pm0.7}$        |$24.3_{\pm0.3}$   |$23.9_{\pm0.5}$      | $25.2_{\pm0.2}$    | $24.8_{\pm0.4}$    |  $\textbf{26.1}_{\pm0.4}$ |
> |                          | 10     | $25.2_{\pm0.3}$    | $24.7_{\pm0.4}$   | $29.7_{\pm0.3}$    |   $29.1_{\pm0.4}$       | $39.7_{\pm0.4}$   | $38.8_{\pm0.7}$  | $\textbf{43.4}_{\pm0.3}$ | $41.7_{\pm0.3}$ |  $43.1_{\pm0.5}$         |
> |                          | 50     | -                             |-                                | $43.6_{\pm0.4}$   |   $42.8_{\pm0.4}$      | $42.9_{\pm0.5}$   | $41.8_{\pm0.5}$  | $50.7_{\pm0.3}$    | $47.9_{\pm0.3}$    |  $\textbf{50.9}_{\pm0.5}$ |
> | TinyImageNet | 1        | -                             | -                               | $3.9_{\pm0.2}$      | $3.6_{\pm0.2}$          | $8.8_{\pm0.3}$    | $8.6_{\pm0.2}$     | $10.4_{\pm0.3}$    | -                  |  $\textbf{10.9}_{\pm0.2}$ |
> |                          | 10     | -                             | -                               | $12.9_{\pm0.4}$    | $12.7_{\pm0.2}$         |$23.2_{\pm0.2}$    |$22.9_{\pm0.5}$.   | $24.5_{\pm0.2}$    | -                  |  $\textbf{25.4}_{\pm0.4}$ |
>
>
> From the results in the table above, we can notice that DC and DM perform stability but still fail to outperform trajectory-based algorithms including our SATM.
>
> ----------- to continue -------------

---

> ### Author Response · Authors · 2024-11-24
>
> ###  ***Questions 1 and 2: motivation and contribution***
> The direct application of existing sharpness-aware strategies is certainly beneficial for performance gains, and we have discussed this in Table 4 of our submission. These optimizers are designed for uni-level optimization problems and do not account for the unrolling cost in the inner loop of bilevel optimization and the inaccurate approximation of the sharpness caused by the complicated inner loop, while also introducing several hyperparameters that increase the computational burden of the learning process.
>
> In our work, we aim to achieve outer loop generalization through an efficient optimizer. With the goal of maintaining a "plug-and-play" approach and avoiding overcomplicating our algorithm, especially in light of the notorious optimization cost introduced by sharpness-aware optimizers, we designed the simple and efficient SATM, which incorporates truncated gradient steps, trajectory reusing, and smoothing of the sharpness approximation.
>
> Regarding the mention of Bayesian Optimization by the reviewer, we believe this might be a typo, as it is not part of our method. Alternatively, if the reviewer could clarify this point, we would be happy to discuss it further.
>
> ###  ***Question 3: research direction***
> We agree with the reviewer that parameterizing the learnable dataset in a more efficient design is a crucial topic in dataset condensation, and it certainly brings performance benefits. However, we also believe that maintaining diversity in research is a valuable contribution in itself. For example, NTK is applied to reformulate the bilevel optimization problem into a uni-level one [7, 8, 9], while DATM approaches dataset condensation from a curriculum learning perspective.
>
> ###  ***Question 4: reference of the experiment results***
> We clarify that the results for DC and DM are cited from the FTD paper, as they use the exact same experimental settings. We appreciate the reviewer for highlighting this missing hyperparameter, which has now been added to ***Appendix A.6*** and also added the learning rate for learning the step size.
>
> [1] Cazenavette, G., Wang, T., Torralba, A., Efros, A.A. and Zhu, J.Y. Dataset distillation by matching training trajectories. In Proceedings of the IEEE/CVF Conference on Computer Vision and Pattern Recognition, 2022.
>
> [2] Guo Z, Wang K, Cazenavette G, LI H, Zhang K, You Y. Towards Lossless Dataset Distillation via Difficulty-Aligned Trajectory Matching. In The Twelfth International Conference on Learning Representations, 2024.
>
> [3] Cui J, Wang R, Si S, Hsieh CJ. Scaling up dataset distillation to imagenet-1k with constant memory. In International Conference on Machine Learning, 2023.
>
> [4] Du, J., Jiang, Y., Tan, V.Y., Zhou, J.T. and Li, H. Minimizing the accumulated trajectory error to improve dataset distillation. In Proceedings of the IEEE/CVF conference on computer vision and pattern recognition, 2023.
>
> [5] Zhao, B., Mopuri, K.R. and Bilen, H., Dataset Condensation with Gradient Matching. In International Conference on Learning Representations, 2021.
>
> [6] Zhao, B. and Bilen, H., Dataset condensation with differentiable siamese augmentation. In International Conference on Machine Learning, 2021.
>
> [7] Nguyen, T., Novak, R., Xiao, L. and Lee, J., Dataset distillation with infinitely wide convolutional networks. Advances in Neural Information Processing Systems, 2021.
>
> [8] Nguyen, T., Chen, Z. and Lee, J., Dataset Meta-Learning from Kernel Ridge-Regression. In International Conference on Learning Representations, 2021.
>
> [9] Loo, N., Hasani, R., Lechner, M. and Rus, D., Dataset distillation with convexified implicit gradients. In International Conference on Machine Learning, 2023.

---

> ### Author Response · Authors · 2024-11-26
>
> Dear reviewer,
>
> Thank you for sharing your concerns regarding our submission. We have addressed your questions and clarified the issues related to experiment result citations. With the extension of the rebuttal period, we are confident there is sufficient time to address your comments in detail. We kindly request your feedback on our rebuttal at your earliest convenience.
>
> Best regards,
> Authors

---

### Official Review · Reviewer_2fvW · 2024-11-09

**Soundness:** 3
**Presentation:** 4
**Contribution:** 3
**Rating:** 6
**Confidence:** 3

**Summary:**

The paper introduces Sharpness-Aware Trajectory Matching (SATM) as an approach to improve dataset condensation. The goal is to synthesize representative samples from large datasets, reducing both computational costs and training time. SATM enhances generalization by minimizing the sharpness in the outer loop of bilevel optimization and uses efficient hypergradient approximation techniques to control memory and computational costs. The method outperforms existing condensation approaches across in-domain and out-of-domain benchmarks, particularly when used in conjunction with sharpness-aware minimization methods.

**Strengths:**

1. SATM’s use of sharpness-aware trajectory matching addresses generalization issues in previous dataset condensation methods and introduces a theoretically sound approach to managing computational complexity.

2. The proposed hypergradient approximation strategies reduce computational overhead, making SATM adaptable and efficient.
3. SATM demonstrates robust performance gains over other state-of-the-art methods across various benchmarks and settings.
4. SATM is compatible with other sharpness-aware optimizers, making it adaptable to a wide range of machine-learning tasks.

**Weaknesses:**

1. **Proposition 3.1**: Could the authors clarify whether $\alpha J$ is greater than or less than 1? If $\alpha J > 1$, the inequality appears incorrect since the norm cannot be bounded by a negative value on the right-hand side. If $\alpha J < 1$, a smaller $\iota$ would yield a tighter bound. Therefore, a discussion on the trade-off between the number of truncated steps and performance is needed. An ablation study on this trade-off would add valuable insights.

2. **Theorem 3.2**: $\theta$ should be treated as a vector, so $\Delta \theta_\tau$ is also a vector, not a scalar. Additionally, the bound on $||\Delta \theta_\tau||$ makes more sense to me, and the upper bound of $\Delta \theta_\tau$ does not necessarily imply that $\theta_\tau$ is close to $\hat{\theta}_\tau$.

3. **Sharpness-Aware Minimization**: Sharpness-aware minimization is known to impose a significant computational burden. The authors argue that minimizing sharpness enhances generalization, but could they elaborate on why sharpness minimization is particularly beneficial in the context of dataset condensation? Further insights on this aspect would strengthen the motivation behind using sharpness-aware methods here.

4. **Formatting Issues**: Ensure there is a space between "min" and $\mathcal{L}$; additionally, Figure 1 currently displays an incomplete label for "training iteration."

**Questions:**

See weakness

---

> ### Author Response · Authors · 2024-11-21
> **Authers' Response**
>
> ###  ***Weakness 1:  $\alpha J $ assumption  and ablation study***
> To ensure the inequality holds, we assume $\alpha J < 1$. Additionally, the learning rate, $\alpha$, is set to 0.01 in practice. We agree that illustrating the truncated steps is closely tied to the model's performance. Below, we provide experimental results for demonstration. In general, the approximation error tends to increase as the number of truncated steps grows.
>
> We chose the settings that require long inner loops for dataset learning. The table below details the experimental settings, including the dataset, the number of images per category (IPC), and the inner loop steps $N$. For example, "CIFAR-10 (1 IPC, 50 steps)" refers to condensing one synthetic image per category with 50 inner loop steps. To analyze the effect on performance, we retained the last $\frac{1}{k}$ steps, where $k =2,3,4,5,6$, of the total inner loop steps. For simplicity, the inner loop steps remained for the first round of hypergradient computation and trajectory reusing in the second round is kept the same which is applied across all experiments. The operation $int(\frac{N}{k})$ is used to determine the remaining inner loop steps. We examined how accuracy changes with the remaining inner loop steps by executing SATM for a fixed number of training iterations (10,000). A clear trend emerged: performance improves as the number of truncated iterations decreases and converges once the differentiation steps reach a certain threshold.
>
> ***Accuracy (%) change with the remaining inner loop step***
>
> |   setting\steps             |  1/6    |  1/5  | 1/4   | 1/3  | 1/2   |
> |-----------------------------|---------|-------|-------|------|-------|
> | CIFAR-10  (1IPC,  50step)   |  45.2   |  48.8 | 47.5  | 49.0 | 49.2  |
> | CIFAR-100 (50IPC, 80step)   |  23.4   |  33.4 | 48.7  | 50.9 | 50.5  |
>
> ###  ***Weakness 2: updated theorem for $||\Delta \theta_{\tau} ||$***
> Thank you for pointing this out. We agree that studying the distance between specific points from two different trajectories is more meaningful. Accordingly, we have updated the theorem and its corresponding proof. The revised theorem is included below, and the proof with some modifications has been updated in the new version of the submission, now available in the Appendix. We are grateful for this suggestion, as we believe it significantly enhances the quality of the paper.
>
>
> ***Theorem 3.2 .***
> Let $\mathcal{L}(\phi, \theta)$ be a function that is $\sigma$-smooth and continuous with respect to its arguments $\phi$ and $\theta$. Additionally, let the second-order derivatives $\nabla_{\phi} \nabla_{\theta} \mathcal{L}(\phi, \theta)$ be $\beta$-continuous. Consider two trajectories obtained by conducting gradient descent training on the datasets $\phi$ and $\phi + \epsilon$, respectively, with a carefully chosen learning rate $\alpha$ and identical initializations. After $\tau$ steps of training, let
>
> $$\Delta \theta_{\tau} = \hat{\theta}_{\tau} (\phi + \epsilon) - \theta_{\tau} (\phi)$$
> Then, we have:
> $$
> ||\Delta \theta_{\tau} || \leq \alpha \tau (2\sigma + \beta \rho).
> $$
>
> ###  ***Weakness 3: the benefits of the flat loss landscape***
> The outer loop in dataset condensation focuses on optimizing the synthetic dataset to achieve better learning performance on downstream tasks. Incorporating sharpness-aware minimization into this process helps guide the optimization towards flatter loss landscapes, which are empirically associated with improved generalization. While sharpness-aware optimization theory remains underexplored in bilevel optimization settings, the PAC-Bayesian generalization bounds for such problems are not yet well developed. Instead, we study this evaluate dataset condensation with trajectory matching on a variety of sharpness-aware optimisers. As demonstrated in Table 4 in our submission, directly applying existing sharpness-aware strategies, such as SAM [1], GSAM [2] and ASAM [3], consistently enhances performance compared to the main baseline, MTT [4]. This highlights the advance of the flat loss landscape to improve dataset condensation by fostering generalization.
>
> ###  ***Weakness 4: editing issues***
> Thank you for highlighting these points. We have addressed the editing issues and made the necessary
> ***Reference***
>
> [1] Foret, Pierre, et al, Sharpness-aware Minimization for Efficiently Improving Generalization. In International Conference on Learning Representations, 2021.
> [2] Zhuang, Juntang, et al. Surrogate Gap Minimization Improves Sharpness-Aware Training. In International Conference on Learning Representations, 2022.
> [3] Kwon, Jungmin, et al. Asam: Adaptive sharpness-aware minimization for scale-invariant learning of deep neural networks. In International Conference on Machine Learning, 2021.
> [4] Cazenavette, George, et al. Dataset distillation by matching training trajectories. In Proceedings of the IEEE/CVF Conference on Computer Vision and Pattern Recognition, 2022.

---

> > ### Comment · Reviewer_2fvW · 2024-11-23
> >
> > Thank you for the authors' response. I will maintain my support for this paper and its acceptance.

---

> > > ### Author Response · Authors · 2024-11-24
> > >
> > > We sincerely appreciate the reviewer’s support for the acceptance of our paper. Please feel free to reach out with any additional questions or concerns. We are more than willing to make further improvements to our work.

---

### Official Review · Reviewer_QXyW · 2024-11-11

**Soundness:** 3
**Presentation:** 3
**Contribution:** 3
**Rating:** 6
**Confidence:** 2

**Summary:**

This paper enhances the generalization capability of dataset condensation by minimizing sharpness in the outer loop of bilevel optimization. Specifically, it introduces Sharpness-Aware Trajectory Matching (SATM), a variant of trajectory matching. SATM jointly minimizes the sharpness and the distance between training trajectories with a tailored loss landscape smoothing strategy. This paper also introduces some techniques to tackle the problems of using SATM, such as the computational overhead, redundancy of the (hyper) gradient calculation, and hyperparameter tuning.
According to the experiments, it achieves the best results in the conventional dataset condensation setting and also shows impressive results in the OOD setting, demonstrating great generalization capability.

**Strengths:**

1. Generalization capability is a good research topic in dataset condensation, and a few papers study it.
2. The overall writing and presentation are satisfactory.
3. SATM and some techniques to tackle the problems when using SATM are novel.

**Weaknesses:**

Missing comparisons with SOTA methods, such as DATM, RDED, and CUDD.
Towards Lossless Dataset Distillation via Difficulty-Aligned Trajectory Matching
On the Diversity and Realism of Distilled Dataset: An Efficient Dataset Distillation Paradigm
Curriculum Dataset Distillation

**Questions:**

How about the results on ResNet-101 and ImageNet-1K.

---

> ### Author Response · Authors · 2024-11-21
> **Authers' Response**
>
> ###  ***Weakness 1:  comparison with DATM and the discussion about RDED***
> ***DATM:*** We thank the reviewers for pointing this out. DATM [1] primarily leverages the difficulty of training trajectories to implement a curriculum-learning-based dataset condensation protocol. While this approach is highly relevant, it is tangential to research lines focusing on optimization efficiency and generalization, such as Tesla [2], FTD [3], and SATM, which emphasize improving gradient approximation and optimization efficiency. Moreover, from an implementation perspective, DATM operates by feeding expert trajectories in an easy-to-hard manner into FTD. In contrast, our work emphasizes the flatness of the loss landscape of the learning dataset from the perspective of the outer loop in bilevel optimization, rather than engaging extensively in direct performance comparisons. That said, we believe our method is compatible with DATM. To support this, we conducted experiments combining the DATM with SATM, denoted by SATM-DA. The results are presented below:
>
> |                 |  IPC   | MTT          |     FTD          |    DATM             |    SATM-DA           |
> |-----------------|--------|--------------|------------------|---------------------|----------------------|
> |                 |  1     | 46.2$\pm$0.8 |  46.8$\pm$0.3    |  46.9$\pm$0.5       | **48.6**$\pm$0.4     |
> | CIFAR-10        |  10    | 65.4$\pm$0.7 |  66.6$\pm$0.3    |  66.8$\pm$0.2       | **68.1**$\pm$0.3     |
> |                 |  50    | 71.6$\pm$0.2 |  73.8$\pm$0.2    |  76.1$\pm$0.3       | **76.4**$\pm$0.6     |
> |                 |  1     | 24.3$\pm$0.3 |  25.2$\pm$0.2    |  27.9$\pm$0.2       | **28.2**$\pm$0.8     |
> | CIFAR-100       |  10    | 39.7$\pm$0.4 |  43.4$\pm$0.3    |  47.2$\pm$0.4       | **48.3**$\pm$0.4     |
> |                 |  50    | 47.7$\pm$0.2 |  50.7$\pm$0.3    |   55.0$\pm$0.2      | **55.7**$\pm$0.3     |
> | Tiny-ImageNet   |  1     |  8.8$\pm$0.3 |  10.4$\pm$0.3    |  **17.1**$\pm$0.3   | 16.4$\pm$0.4         |
> |                 |  10    | 23.2$\pm$0.1 |  24.5$\pm$0.2    |  31.1$\pm$0.3       | **32.3**$\pm$0.6     |
>
> From the results, we observe that SATM is compatible with DATM and can directly benefit from the easy-to-hard training protocol. We thank the reviewer for pointing this out, and we will incorporate this content into our manuscript.
>
> ***RDED:*** We appreciate the reviewer for highlighting this paper. RDED [4] has indeed introduced new perspectives to the dataset distillation field by constructing synthetic images from original image crops and labelling them with a pre-trained model. However, our work falls within the training trajectory matching area and focuses on efficient bilevel optimization with a long inner loop. Our goal is to enhance the generalization ability of synthetic data by developing an efficient, sharpness-aware optimizer for bilevel optimization. The proposed SATM reduces computational costs through a straightforward, mathematically supported approximation of the existing sharpness-aware optimizer with double unrolling.
>
> We agree that RDED brings superior performance to dataset condensation and outperforms all existing trajectory-matching-based methods. We certainly include a discussion of this paper in our related work section.
>
> ----------- to continue -------------

---

> ### Author Response · Authors · 2024-11-24
>
> ### ***Question 1: generalisation to ResNet-101 and ImageNet 1K***
> Testing the algorithm in a scaled-up setting is indeed a valuable suggestion. However, in the context of trajectory matching [1,2,3,4] and neural tangent kernel-based algorithms [5], using ResNet-101 presents challenges due to the long inner loop unrolling and the size of the neural tangent kernel. Additionally, storing a collection of ResNet-101 training trajectories as expert trajectories can be computationally expensive.
>
> We adopt Tesla's evaluation protocol and test SATM on ImageNet-1K to showcase its efficiency in training memory and time, as well as its ability to generalize in scaled-up settings. To ensure a fair comparison, SATM utilizes the same soft labels as Tesla. As demonstrated below, SATM effectively handles the complex task of dataset condensation and achieves superior performance compared to Tesla. The results have been updated in Appendix A.10 in the latest version.
>
> |    Dataset    | IPC |    Tesla          |        SATM              |
> |---------------|-----|-------------------|--------------------------|
> |  ImageNet-1K  |  1  |   7.7$\pm$0.2     |    **8.2**$\pm$0.4       |
> |               |  2  |  10.5$\pm$0.3     |    **11.4**$\pm$0.2      |
> |               | 10  |  17.8$\pm$1.3     |    **18.5**$\pm$0.9      |
> |               | 50  |  27.9$\pm$1.2     |    **28.4**$\pm$1.1      |
>
>
> ***Reference***
>
> [1] Guo Z, Wang K, Cazenavette G, LI H, Zhang K, You Y. Towards Lossless Dataset Distillation via Difficulty-Aligned Trajectory Matching. In The Twelfth International Conference on Learning Representations, 2024.
>
> [2] Cui J, Wang R, Si S, Hsieh CJ. Scaling up dataset distillation to imagenet-1K with constant memory. In International Conference on Machine Learning, 2023.
>
> [3] Du, J., Jiang, Y., Tan, V.Y., Zhou, J.T. and Li, H. Minimizing the accumulated trajectory error to improve dataset distillation. In Proceedings of the IEEE/CVF conference on computer vision and pattern recognition, 2023.
>
> [4] Sun, P., Shi, B., Yu, D. and Lin, T.. On the diversity and realism of distilled dataset: An efficient dataset distillation paradigm. In Proceedings of the IEEE/CVF Conference on Computer Vision and Pattern Recognition, 2024.
>
> [5] Nguyen, T., Novak, R., Xiao, L. and Lee, J. Dataset distillation with infinitely wide convolutional networks. Advances in Neural Information Processing Systems, 2021.

---

### Author Response · Authors · 2024-11-21
**To All Reviewers**

We would like to express our gratitude to all the reviewers for their time and insightful feedback. We also appreciate their recognition of the solid foundation of our proposed algorithm, SATM, which is supported by thorough analysis. The primary concern raised pertains to some of the experimental results. In response, we have conducted additional experiments to further demonstrate the efficiency of our algorithm and have addressed each specific question raised by the reviewers. The latest submission has been updated accordingly, with all these changes included in the appendix section.

---

### Meta-Review · Area_Chair_XJs6 · 2024-12-25

**Metareview:**

**Summary:** This paper introduces Sharpness-Aware Trajectory Matching (SATM), which aims to improve dataset condensation by focusing on the flatness of the loss landscape in bilevel optimization. The authors introduce several techniques, such as hypergradient approximation and sharpness-aware optimization, to address computational challenges and enhance generalization. SATM is evaluated on standard and out-of-domain benchmarks, demonstrating improvements over prior methods like MTT and Tesla.

**Decision:** This is a borderline submission, and I have carefully read the manuscript along with the reviewers' comments and discussions. While there are some concerns regarding the empirical validation and the marginal improvements over the state-of-the-art (SOTA), my primary concern lies with the novelty of the work, an issue also raised by several reviewers. Specifically, the study appears to be a combination of existing techniques — introducing SAM to BO and applying it to DC. Although the overall workflow is reasonable, I believe this type of incremental contribution does not meet ICLR's acceptance standards.

Some reviewers highlighted the theoretical analysis as a strength of the paper, but I hold a different perspective. Proposition 3.1, while relevant, follows directly from existing work (as acknowledged by the authors), and both the statement and proof of Theorem 3.2 adhere to a standard and incremental approach. More critically, the proof lacks rigor, as it relies on a first-order Taylor expansion (if my understanding is correct), rendering the inequality non-strict.

Given these limitations, I find that the shortcomings outweigh the contributions, despite the relevance of the problem and the potential of the approach. Therefore, I recommend rejecting this submission.

**Additional Comments On Reviewer Discussion:**

During the rebuttal period, the authors provided additional experiments to address the reviewers' concerns, including memory and time costs, comparisons with DATM, and evaluations on ImageNet-1K. While the rebuttal clarified certain points, it did not sufficiently alleviate several major concerns raised by the reviewers. Additionally, given the borderline nature of this submission and the limited feedback from reviewers, I carefully reviewed the paper myself and shared the reviewers' concerns regarding its novelty. Additionally, I find the theoretical contributions of the paper to be relatively incremental, and some of the proofs lack rigor.

---

### Decision · Program_Chairs · 2025-01-22

Reject